# Midway Network: Learning Representations for Recognition and Motion from Latent Dynamics

**Chris Hoang, Mengye Ren**
New York University
{ch3451, mengye}@nyu.edu

## Abstract

Object recognition and motion understanding are key components of perception that complement each other. While self-supervised learning methods have shown promise in their ability to learn from unlabeled data, they have primarily focused on obtaining rich representations for either recognition or motion rather than both in tandem. On the other hand, latent dynamics modeling has been used in decision making to learn latent representations of observations and their transformations over time for control and planning tasks. In this work, we present Midway Network, a new self-supervised learning architecture that is the first to learn strong visual representations for both object recognition and motion understanding solely from natural videos, by extending latent dynamics modeling to this domain. Midway Network leverages a *midway* top-down path to infer motion latents between video frames, as well as a dense forward prediction objective and hierarchical structure to tackle the complex, multi-object scenes of natural videos. We demonstrate that after pretraining on two large-scale natural video datasets, Midway Network achieves strong performance on both semantic segmentation and optical flow tasks relative to prior self-supervised learning methods. We also show that Midway Network's learned dynamics can capture high-level correspondence via a novel analysis method based on forward feature perturbation. Code is provided at this link.

## 1 Introduction

Animals and humans are able to recognize objects and predict their motion by observing the dynamic world with little to no supervision. Inspired by this capability, research in deep learning has made significant progress in emulating "learning by observing." Prior work has shown that observing objects through time via video streams can serve as a powerful learning signal (Földiák, 1991; Wiskott & Sejnowski, 2002; Wang & Gupta, 2015; Srivastava et al., 2015). Others have shown that self-supervised learning (SSL) methods can learn strong visual representations from vast amounts of unlabeled data (Goyal et al., 2022; Oquab et al., 2023; Fan et al., 2025).

Among a number of perception abilities attained via observation, object recognition and motion understanding are two intertwined core components. Recognition allows one to identify the same object across views to establish correspondence; conversely, motion links the same object across spacetime to enable learning of its invariant properties (Simonyan & Zisserman, 2014; Xu & Wang, 2021). However, most prior work on visual SSL has focused on learning representations for *either* object recognition or motion understanding, but not both in tandem. Image SSL methods (Chen et al., 2020b; He et al., 2020; Grill et al., 2020; Caron et al., 2021; Assran et al., 2023) have demonstrated strong capabilities in learning semantic representations, but most operate on iconic, i.e., single-subject, image datasets which are human-curated and additionally lack temporal information to learn motion. More recently, some have proposed performing SSL on natural videos, which depict real-world scenes and can approximate the observational perspective of animals. Nonetheless, these methods either do not utilize motion transformations for training (Gordon et al., 2020; Venkataramanan et al., 2024) or rely on external optical flow networks to incorporate motion as a learning signal (Xiong et al., 2021; Wang et al., 2025). On the other hand, self-supervised methods that focus on learning motion as a pixel-correspondence (Liu et al., 2019; Jonschkowski et al., 2020; Luo et al., 2021; Stone et al., 2021) or cross-view reconstruction task (Weinzaepfel et al., 2023) result in poor semantic representations. Only a few works aim to learn both semantic and motion features (Bardes et al.,

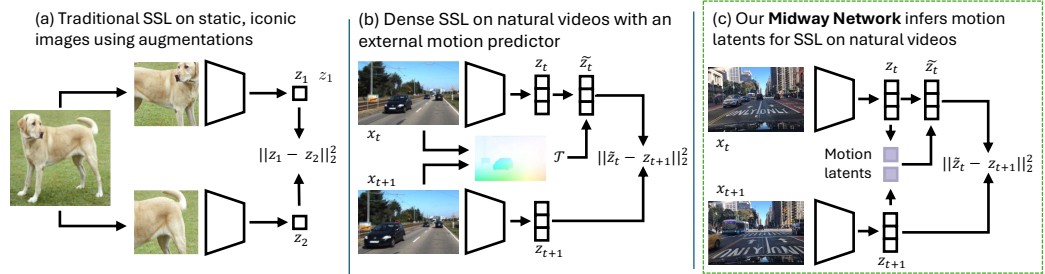

Figure 1: (a) Traditional SSL methods focus on learning representations for object recognition and lean on curated, iconic image data for training. (b) Dense SSL methods extend training to natural videos, but either do not utilize motion transformations (Gordon et al., 2020; Venkataramanan et al., 2024) for training or rely on external networks to incorporate motion (Xiong et al., 2021; Wang et al., 2025). (c) Our proposed Midway Network jointly learns representations of semantics and motion from solely natural videos via latent dynamics modeling. The learned image-level representations can be used towards downstream object recognition and motion understanding tasks.

2023), but these methods still depend on curated, iconic image data for training. How can we jointly learn rich representations for object recognition and motion understanding solely from natural videos?

Theories in neuroscience have proposed that animals use internal inverse and forward dynamics models and future sensory prediction, i.e., predictive coding, to perform motor control, planning, and perception (Shidara et al., 1993; Miall & Wolpert, 1996; Wolpert et al., 1998; Rao & Ballard, 1999; Friston, 2005). Works in decision making have also suggested using latent dynamics modeling for representation learning, but focus on control and planning tasks in simulated and lab environments (Brandfonbrener et al., 2023; Schmidt & Jiang, 2024; Cui et al., 2024). Together, these studies point to latent dynamics modeling as a natural mechanism for learning useful representations of visual observations and their transformations over time, e.g., motion.

Building on this observation, we propose **Midway Network**, a new SSL architecture that is the first to learn both recognition and motion understanding solely from natural videos, by extending latent dynamics modeling to this domain. Midway Network is centered around a *midway* top-down path, which infers motion latents between video frames via inverse dynamics that are subsequently used to condition the forward predictions. We rely on two design choices in order to better model the complex, multi-object scenes in natural videos. First, we formulate the forward prediction objective over *dense* features, rather than global features like in previous works (Cui et al., 2024). Second, Midway Network introduces a hierarchical architecture with backward and lateral layers to refine the motion latents and representations over multiple feature levels, inspired by optical flow networks (Sun et al., 2018).

Midway Network shows strong capability of learning image-level representations for object recognition and motion understanding after pretraining on large-scale natural video datasets. In particular, Midway Network outperforms prior SSL methods on optical flow tasks while also achieving competitive performance on semantic segmentation tasks for both BDD100K (Yu et al., 2020) and Walking Tours (Venkataramanan et al., 2024) pretraining. We also show that Midway Network's dynamics models can capture high-level correspondence, supported by evidence from our novel analysis method based on forwarded feature perturbation. Finally, our ablation studies demonstrate that our hierarchical design components are important for downstream performance.

To summarize, our contributions are:

- We present Midway Network, a novel SSL architecture that is the first to learn rich image-level representations for object recognition and motion understanding solely from natural videos. It leverages a dense forward prediction objective and hierarchical design to better capture the complexity of natural videos.
- We show that Midway Network achieves strong performance on *both* optical flow (FlyingThings, MPI-Sintel) and semantic segmentation (BDD100K, CityScapes, WT-Sem, ADE20K) when pretrained on natural video datasets, compared to prior SSL baselines which only attain substantial performance in one of the two tasks.
- We demonstrate Midway Network's ability to capture high-level correspondences between video frames with evidence from our novel analysis method based on forwarded feature perturbation.

## 2 RELATED WORK

**Predictive modeling.** This work builds upon research in predictive modeling from neuroscience and deep learning. Many works in neuroscience have explored predictive coding, a theory positing how the brain continuously predicts future sensory inputs with hierarchical networks to perform perception (Rao & Ballard, 1999; Rao & Sejnowski, 1999; Lee & Mumford, 2003; Friston, 2005; Summerfield et al., 2006). In particular, Friston's theory (Friston, 2005) describes how perception may be split into *recognition*, inferring causes of sensory inputs, which is reminiscent of representation learning and inverse dynamics, and *generation*, predicting (future) sensory inputs from causes, which is akin to forward dynamics. Biological evidence of predictive coding has also been found, such as in monkey neural cells after receptive field excitation (Livingstone, 1998) and in functional magnetic resonance imaging data of human subjects following visual stimuli under varying expectation levels (Egner et al., 2010). In deep learning, prior works such as PredNet (Chalasani & Principe, 2013; Lotter et al., 2017) have proposed architectures inspired by predictive coding to perform video prediction. More generally, there has been a line of research in leveraging prediction of future frames in videos as a learning objective (Softky, 1995; Finn et al., 2016; Villegas et al., 2018; Feichtenhofer et al., 2022). Others have developed predictive modeling methods that perform video prediction in latent feature space (Vondrick et al., 2016; Bardes et al., 2024). Midway Network is inspired by these ideas, extending dynamics-conditioned predictive modeling to natural videos with a new hierarchical architecture in order to learn rich representations for object recognition and motion understanding.

**Dynamics modeling.** Prior works have suggested that animals use internal inverse and forward dynamics models for motor control and planning (Wolpert et al., 1995; Miall & Wolpert, 1996; Flanagan & Wing, 1997; Wolpert et al., 1998; Kitazawa et al., 1998; Jordan & Rumelhart, 2013). Inverse and forward dynamics have subsequently been used in works like DynaMo (Cui et al., 2024) to learn latent visual and action representations for robotic manipulation and control tasks (Brandfonbrener et al., 2023; Chen et al., 2024; Ye et al., 2025), but they have only focused on simulated or controlled environments. World models are a concurrent line of work which learn a latent dynamics model of the environment to enable efficient policy learning and long-horizon planning, but prior works such as Dreamer and V-JEPA 2 have relied on ground truth reward signals or actions (Ha & Schmidhuber, 2018; Hafner et al., 2019; 2020; Schwarzer et al., 2021; Hu et al., 2023; Assran et al., 2025). In particular, DINO-WM (Zhou et al., 2024) proposed training a forward dynamics predictor over DINOv2 (Oquab et al., 2023) patch features, but this method also required access to ground truth actions. More recently, generative models, such as the Genie series, have emerged as a promising approach for learning world models and interactive environments (Menapace et al., 2022; Yang et al., 2024; Parker-Holder et al., 2024; Sun et al., 2024). Midway Network utilizes inverse and forward dynamics to tackle a new problem: learning rich image-level representations for recognition and motion understanding solely from natural videos. It leverages *dense* forward prediction and a new hierarchical refinement architecture to capture the complex, multi-object scenes in this data domain.

**Visual self-supervised learning.** SSL on visual data has enjoyed a long history, from denoising autoencoders (Vincent et al., 2010; Pathak et al., 2016; Chen et al., 2020a; He et al., 2022) to joint embedding (Grill et al., 2020; Chen et al., 2020b; He et al., 2020; Caron et al., 2021; Bardes et al., 2022) and joint-embedding predictive (Assran et al., 2023; Garrido et al., 2024) models. These works primarily aim to learn semantic representations from iconic, single-subject images. Following their success, others have proposed methods to learn from dense, multi-subject images by leveraging losses on local features (Wang et al., 2021; Xie et al., 2021; Bardes et al., 2022; Zhang et al., 2023). While prior work uses motion from natural videos to learn visual representations (Xiong et al., 2021; Wang et al., 2025), these approaches either rely on external supervised flow networks or use motion only to construct training views (Jabri et al., 2020; Gordon et al., 2020; Venkataramanan et al., 2024). In contrast, our work also jointly learns representations of the motion transformations themselves. A separate line of work focuses on learning motion as a cross-view pixel correspondence (Liu et al., 2019; Jonschkowski et al., 2020; Luo et al., 2021; Stone et al., 2021) or reconstruction task (Weinzaepfel et al., 2022; 2023); however, the resulting features have poor recognition capability. Video SSL methods (Tong et al., 2022; Wei et al., 2022; Bardes et al., 2024) tackle learning clip-level representations for action recognition tasks, whereas Midway Network and our baselines target image-level representations. While a few video SSL works (Qing et al., 2022) also explore hierarchical designs for learning, these hierarchies are only related to the temporal structure of videos for sampling training pairs. Finally, MC-JEPA seeks to learn both semantic and motion features (Bardes et al., 2023), but unlike Midway Network, it still relies on curated, iconic image data for training.

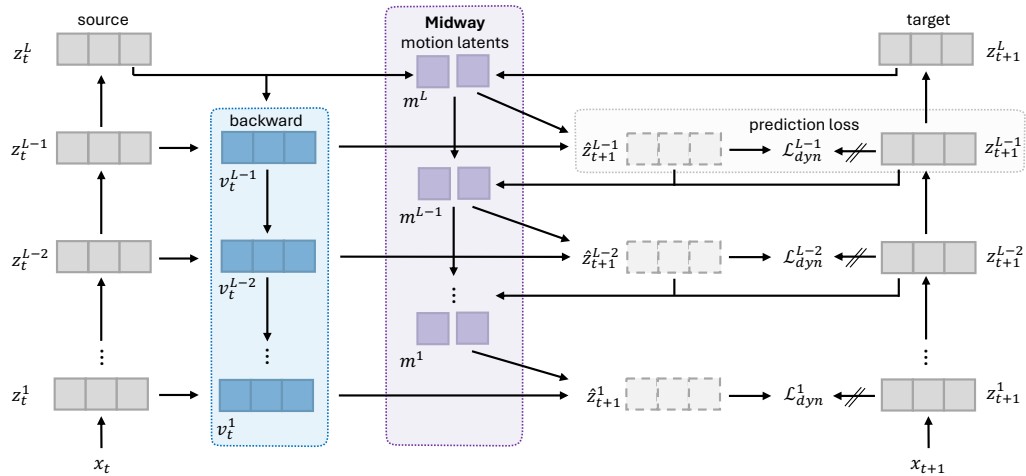

Figure 2: Midway Network employs a hierarchical design in which the *midway* path infers motion latents $m$ between source and target features in a top-down manner. Within each level of this hierarchy, backward layers with top-down and lateral connections refine the source features $z_t^l$. Forward prediction blocks, conditioned on the refined features $v_t^l$ and motion latents $m^{l+1}$, predict the dense target features $z_{t+1}^l$, and the prediction loss $\mathcal{L}_{dyn}$ jointly trains all components at each level.

## 3  MIDWAY NETWORK

We present Midway Network, a new SSL architecture that uses latent dynamics modeling to learn representations for object recognition and motion understanding solely from natural videos. At the heart of Midway Network is a *midway* path that infers motion latents to describe the transformation between a source and target video frame. The visual encoder extracts features from the raw video frames, and the backward layers refine these features with lower-level information in a top-down manner. The forward dynamics model, conditioned on the source frame backward features and motion latents, predicts the *dense* target frame features, and the resulting prediction error is used to jointly train all components of the model. Midway Network employs a hierarchical design, where the forward prediction objective is placed at multiple feature levels, and the forward predictions from higher feature levels are used as the input to refine the motion latents at lower levels. The architecture is illustrated in Figure 2, and the computations for the dense forward prediction objective are summarized in Algorithm 1.

**Preliminaries.** The model inputs are pairs of source and target video frames, $x_t$ and $x_{t+1}$. Following the SSL knowledge distillation paradigm (Grill et al., 2020; Caron et al., 2021), we encode the video frames into features using source and target networks, $z_t = f_\theta(x_t)$ and $z_{t+1} = f_{\tilde{\theta}}(x_{t+1})$, where $\tilde{\theta}$ is updated using an exponential moving average of the student parameters $\theta$. Midway Network operates at multiple feature levels, so we use the notation that $z^l$ are the features of the $l$-th level, ordered from lowest level 1 to highest level $L$.

**Algorithm 1** Dense forward prediction objective.

1:  $m^{L+1} \leftarrow 0, v_t^L \leftarrow z_t^L$.
2:  $z_t \leftarrow f_\theta(x_t), z_{t+1} \leftarrow f_{\tilde{\theta}}(x_{t+1}), \hat{z}_{t+1}^L \leftarrow z_t^L$.
3:  **for** $l \leftarrow L-1$ to 1 **do**:
4:      $m^{l+1} \leftarrow \text{midway}(m^{l+2}, \hat{z}_{t+1}^{l+1}, z_{t+1}^{l+1}) + m^{l+2}$.
5:      $v_t^l \leftarrow \text{backward}(z_t^l, v_t^{l+1})$.
6:      $\hat{z}_{t+1}^l \leftarrow \text{predictor}(v_t^l, m^{l+1})$.
7:      $\bar{\hat{z}}_{t+1}^l \leftarrow \frac{\hat{z}_{t+1}^l}{||\hat{z}_{t+1}^l||_2}, \bar{z}_{t+1}^l \leftarrow \frac{z_{t+1}^l}{||z_{t+1}^l||_2}$.
8:      $\mathcal{L}_{dyn}^l \leftarrow ||\bar{\hat{z}}_{t+1}^l - \bar{z}_{t+1}^l||_2^2$.
9:  **end for**
10: $\mathcal{L}_{dyn} \leftarrow \sum\limits_{l=1}^{L-1} \mathcal{L}_{dyn}^l$.

**Motion latents via *midway* path.** The midway path aims to learn motion latents that capture the transformation between observations over time via inverse dynamics. Specifically, the midway inverse dynamics model is a transformer that takes in previous motion latents $m^{l+1}$ and the source and target features $z_t^l$ and $z_{t+1}^l$ as input, and outputs the motion latents $m^l$ for the next level. The motion latents accumulate over levels, i.e. $m^l = \text{midway}(m^{l+1}, z_t^l, z_{t+1}^l) + m^{l+1}$. The initial motion latents are learnable tokens. For every level besides the top level, we use the output of the higher level's forward prediction, $\hat{z}_t^l$, instead of the features $z_t^l$ as input. Thus, the model learns to refine the motion latents in a top-down manner, conditioned on the higher-level predictions. This design is motivated by how prior optical flow methods (Sun et al., 2018; Jonschkowski et al., 2020) would use intermediate flow estimates to warp features before computing cost volumes, which would subsequently be used to refine flow predictions at lower levels.

**Backward features.** Prior works, from Ladder Networks (Valpola, 2015) to PooDLe (Wang et al., 2025), have proposed backward layers with top-down and lateral connections to relieve higher-level features of the burden of encoding low-level details. In this work, backward layers are used to refine features in a top-down manner by using lateral connections to incorporate lower-level information. Specifically, the backward layers are transformer blocks that use cross-attention (Lin et al., 2022), where laterally-connected features $z_t^l$ are used as queries that attend to higher-level backward features $v_t^{l+1}$, which serve as keys and values.

**Dense forward prediction.** The forward dynamics model is also a transformer that takes in backward features $v_t^l$ and motion latents $m^{l+1}$ as input, concatenated along the spatial dimension, and predicts the dense features of the target frame. The dense forward prediction objective is then to minimize the prediction error between the predicted features $\hat{z}_{t+1}^l$ and the realized target features $z_{t+1}^l$. The prediction error is the mean squared error between the normalized dense predictions and targets:

$$\mathcal{L}_{dyn}^l = \|\bar{\hat{z}}_{t+1}^l - \bar{z}_{t+1}^l\|_2^2. \tag{1}$$

*Forward prediction gating.* In a standard transformer block, the input token value is always propagated forward due to the residual connection — this biases the computation towards the identity mapping. However, we would like the forward transformer model to learn whether the object captured by an input token has moved, i.e., if its features can be computed from tokens at *other* spatial locations, rather than defaulting to the identity location. Thus, we introduce learnable gating units for the residual connection in the transformer layers of the forward dynamics model. The gating unit is a multi-layer perceptron that learns a vector-wise gating weight between 0 and 1 for the residual connection of each input token of $v_t$. Specifically, the transformer block is modified with gating unit $g$ such that the input to the feedforward network, $h$, is computed as:

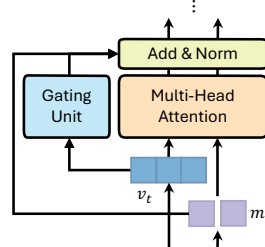

Figure 3: Attention layer with gating unit on $v_t$.

$$h = g(x) \cdot x + \text{Attention}(x). \tag{2}$$

We do not use gating units in the first transformer block to provide sufficient information for initial estimates of attention, nor do we use them for the motion latents $m$ to fully propagate the motion information. In our experiments, we find that the gating units improve semantic feature quality and interpretability of the learned dynamics models, as shown in Section 4.4.

**Invariance objective.** We utilize an additional joint-embedding invariance objective over smaller crops to encourage the visual encoder to learn semantic features, following PooDLe (Wang et al., 2025). In our experiments, we use the DINO (Caron et al., 2021) objective with projection heads on top of the source and target networks. This can be viewed as a form of regularization for the features that are subsequently used in the latent dynamics modeling.

## 4 EXPERIMENTS

We evaluate Midway Network by pretraining on large-scale natural video datasets, BDD100K (Yu et al., 2020) and Walkings Tours (WT) (Venkataramanan et al., 2024), and evaluating the learned image and motion latent representations on downstream semantic segmentation and optical flow tasks. In our experiments, we study whether Midway Network learns good visual features for both object recognition and motion understanding. We further analyze how each component of Midway Network contributes to downstream performance and what information does its dynamics models capture.

### 4.1 SETUP

**Pretraining.** We pretrain Midway Network on two large-scale video datasets from different domains. **BDD100K** (Yu et al., 2020) is a dataset of 100,000 dashcam driving videos collected in varying weather, lighting, and time-of-day conditions from New York and the San Francisco Bay Area. Each video is 40 seconds long at 720p and 30 fps. We pretrain on all 70,000 videos in the train split. **Walking Tours (WT)** (Venkataramanan et al., 2024) is a dataset of 10 first-person YouTube walking videos collected in various cities of Europe and Asia, with outdoor and indoor scenes, and natural transitions in lighting and location. The videos range from 59 minutes to 2 hours 55 minutes, at 720p and 30 fps. We pretrain on the Venice video following DoRA (Venkataramanan et al., 2024)'s original setup.[1]

---

[1]Due to computational constraints, we did not pretrain on all 10 videos.

**Downstream evaluations.** We evaluate Midway Network's pretrained representations on semantic segmentation tasks to gauge object recognition capability. For BDD pretraining, we perform linear and UperNet readout on the BDD and CityScapes (Cordts et al., 2016) benchmarks following FlowE (Xiong et al., 2021). For WT pretraining, we perform UperNet finetuning on the WT-Sem (Wang et al., 2025) and ADE20K (Zhou et al., 2017) benchmarks following DoRA (Venkataramanan et al., 2024) and PooDLe (Wang et al., 2025). For linear readout only, we use the backward layer features following PooDLe. We also evaluate Midway Network on optical flow tasks to assess motion understanding. We follow CroCo v2 (Weinzaepfel et al., 2023)'s finetuning evaluation protocol, replacing their binocular decoder with our midway inverse dynamics and forward dynamics models — baselines without binocular components also use the dynamics models, but with randomly initialized weights. We finetune models pretrained on BDD on TartanAir (Wang et al., 2020), MPI-Sintel (Butler et al., 2012), FlyingThings (Mayer et al., 2016), and FlyingChairs (Dosovitskiy et al., 2015) datasets, and evaluate on the corresponding validation splits of FlyingThings and MPI-Sintel. We report mean intersection-over-union (mIoU) and pixel-level accuracy (Acc) for semantic segmentation, and endpoint error (EPE) for optical flow. More details on evaluation settings are provided in Appendix B.

**Baselines.** We compare Midway Network to iconic image SSL methods (DINO, iBOT (Caron et al., 2021; Zhou et al., 2021)), multi-object SSL methods (DoRA, PooDLe (Venkataramanan et al., 2024; Wang et al., 2025)), and masked reconstruction methods (CroCo v2, VideoMAE, MAE (Weinzaepfel et al., 2023; Tong et al., 2022; He et al., 2022)). DoRA uses 8-frame clips for training, VideoMAE uses 16-frame clips, and iBOT and MAE use single frames. Midway Network and all other baselines learn from pairs of frames. We also implement a modified version of DynaMo (Cui et al., 2024) that uses ViT-S as the encoder and includes the DINO invariance objective. We use official implementations to pretrain the baselines on BDD and WT. All baselines are trained on $224 \times 224$ resolution, except for PooDLe in Table 2, which uses $512 \times 1024$.

**Implementation.** We use ViT-S and ViT-B sized vision transformers for our visual encoders. For the midway inverse dynamics, forward dynamics, and backward models, we use decoder-only transformers (Vaswani et al., 2017), with the backward layers using cross-attention (Lin et al., 2022) blocks. We largely follow the guidelines provided by PooDLe (Wang et al., 2025) on data sampling from natural videos. Specifically, we sample pairs of frames $0.5 \sim 1$ seconds apart, one per video per epoch for BDD, and 0.5 seconds apart, for all possible pairs per epoch for WT-Venice. For the dense forward prediction objective, we sample larger crops of area range $[0.2, 0.4]$ at the same location for both frames. We take smaller initial crops of area range $[0.05, 0.2]$ at the same location for both frames, from which global and local crops are sampled for the DINO joint-embedding objective. All crops are resized to $224 \times 224$ resolution. Appendix B provides more details on implementation, compute resources, and comparisons of training cost across the different methods.

## 4.2 SEMANTIC SEGMENTATION AND OPTICAL FLOW RESULTS

Table 1: Semantic segmentation and optical flow evaluations for BDD100K $224 \times 224$ resolution pretraining. Sem. Seg. is conducted with frozen backbone and optical flow is conducted with finetuning. [†]DynaMo is modified to use a ViT-S encoder and DINO objective.

| Method | Arch | Ep. | BDD100K Sem. Seg. | | | | Cityscapes Sem. Seg. | | | | Optical Flow | | | |
|---|---|---|---|---|---|---|---|---|---|---|---|---|---|---|
| | | | Linear | | UperNet | | Linear | | UperNet | | FlyingThings | | MPI-Sintel | |
| | | | ↑mIoU | ↑Acc | ↑mIoU | ↑Acc | ↑mIoU | ↑Acc | ↑mIoU | ↑Acc | ↓EPE (c) | ↓EPE (f) | ↓EPE (c) | ↓EPE (f) |
| PooDLe (Wang et al., 2025) | R50 | 300 | 35.1 | 87.8 | 47.4 | 91.0 | **44.8** | 89.0 | **59.2** | 93.4 | - | - | - | - |
| iBOT (Zhou et al., 2021) | ViT-S | 800 | 27.2 | 85.4 | 35.5 | 88.7 | 32.0 | 86.2 | 44.0 | 90.3 | 18.5 | 18.0 | 13.0 | 13.7 |
| DINO (Caron et al., 2021) | ViT-S | 300 | 36.7 | 89.3 | 49.3 | 92.0 | 41.5 | 90.4 | 57.9 | 93.3 | 16.8 | 13.8 | 11.5 | 10.8 |
| VideoMAE (Tong et al., 2022) | ViT-S | 300 | 7.8 | 50.3 | 10.9 | 58.6 | 6.4 | 44.9 | 11.7 | 62.9 | 16.2 | 16.1 | 7.2 | 7.6 |
| CroCo v2 (Weinzaepfel et al., 2023) | ViT-S | 300 | 21.2 | 80.0 | 31.9 | 87.0 | 24.0 | 81.5 | 37.5 | 89.0 | 9.7 | 9.4 | 5.1 | 5.8 |
| DoRA (Venkataramanan et al., 2024) | ViT-S | 300 | 30.4 | 87.2 | 40.8 | 90.0 | 36.2 | 88.2 | 51.3 | 91.9 | 16.5 | 15.1 | 11.5 | 11.9 |
| DynaMo[†] (Cui et al., 2024) | ViT-S | 300 | 36.8 | 89.4 | 47.4 | 91.7 | 41.2 | 90.3 | 57.2 | 93.1 | - | - | - | - |
| Midway (enc. only) | ViT-S | 300 | - | - | - | - | - | - | - | - | 16.6 | 13.5 | 11.7 | 10.9 |
| Midway | ViT-S | 300 | **39.7** | **90.3** | **50.4** | **92.4** | 43.0 | **90.9** | 58.5 | **93.5** | **7.3** | **6.8** | **4.1** | **4.9** |
| DINO (Caron et al., 2021) | ViT-B | 300 | 44.0 | 90.9 | 53.8 | 92.7 | 48.5 | 91.7 | **62.7** | **94.2** | 17.4 | 14.8 | 12.1 | 14.1 |
| CroCo v2 (Weinzaepfel et al., 2023) | ViT-B | 300 | 16.3 | 72.4 | 26.5 | 84.4 | 18.2 | 75.0 | 28.9 | 84.6 | **6.1** | **5.8** | **3.0** | **3.8** |
| Midway | ViT-B | 300 | **48.2** | **91.6** | **55.2** | **93.1** | **51.1** | **92.1** | 62.2 | 94.0 | 7.0 | 6.4 | 4.1 | 4.8 |

**BDD100K pretraining.** Table 1 shows results on BDD100K and CityScapes semantic segmentation, and FlyingThings and MPI-Sintel optical flow benchmarks after BDD100K pretraining. Notably, Midway Network is the only model to perform well on both semantic segmentation and optical flow tasks overall. For semantic segmentation, Midway Network outperforms all baselines on BDD100K, and its learned visual features also transfer well to CityScapes, where they are competitive with the best-performing baseline, PooDLe, which relies on an external supervised optical flow network. Note that even without the backward network, our model achieves 39.2 mIoU and 90.1 Acc on BDD100K

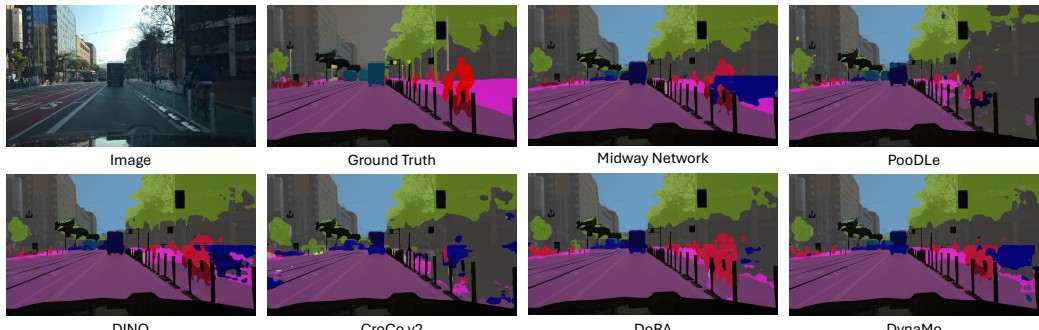

Figure 4: Visualization of BDD semantic segmentation UperNet readout. Midway Network is able to produce cleaner object boundaries, particularly for the cyclist on the right.

Table 2: Semantic segmentation and optical flow evaluations for WT-Venice $224 \times 224$ resolution pretraining. Sem. Seg. and optical flow are conducted with finetuning. [†]PooDLe on $512 \times 1024$ resolution pretraining from their original table (Wang et al., 2025). [*]iBOT results taken from DoRA (Venkataramanan et al., 2024).

| Method | Arch | Ep. | WT-Sem Sem. Seg. UperNet | | ADE20K Sem. Seg. UperNet | | Optical Flow | | | |
| | | | | | | | FlyingThings | | MPI-Sintel | |
| | | | ↑mIoU | ↑Acc | ↑mIoU | ↑Acc | ↓EPE (c) | ↓EPE (f) | ↓EPE (c) | ↓EPE (f) |
|---|---|---|---|---|---|---|---|---|---|---|
| PooDLe[†] (Wang et al., 2025) | R50 | 20 | **13.7** | 85.4 | **36.6** | **77.9** | - | - | - | - |
| iBOT[*] (Zhou et al., 2021) | ViT-S | 100 | - | - | 33.9 | - | - | - | - | - |
| MAE (He et al., 2022) | ViT-S | 100 | 8.9 | 81.5 | 24.1 | 71.4 | 17.6 | 16.4 | 11.1 | 11.8 |
| VideoMAE (Tong et al., 2022) | ViT-S | 100 | 3.3 | 67.9 | 7.8 | 55.6 | 15.9 | 15.8 | 7.0 | 7.4 |
| DINO (Caron et al., 2021) | ViT-S | 100 | 11.0 | 83.0 | 29.2 | 74.7 | 15.5 | 14.0 | 12.4 | 13.8 |
| CroCo v2 (Weinzaepfel et al., 2022) | ViT-S | 100 | 11.3 | 84.4 | 32.0 | 75.7 | 9.6 | 9.1 | 5.9 | **6.4** |
| DoRA (Venkataramanan et al., 2024) | ViT-S | 100 | 13.6 | **85.7** | 35.2 | 77.7 | 17.9 | 13.3 | 12.4 | 12.4 |
| Midway | ViT-S | 100 | 13.1 | 85.4 | 33.4 | 76.9 | **7.7** | **7.4** | **5.2** | 6.6 |

linear readout, continuing to outperform the baselines. Midway Network also surpasses all baselines' performance on FlyingThings and MPI-Sintel optical flow. As shown by *Midway Network (enc. only)*, performance on optical flow drops drastically if we do not initialize the midway inverse and forward dynamics models with the pretrained weights, indicating that the dynamics models have learned features that are useful towards motion estimation. We also demonstrate that Midway Network's downstream performance also scales with larger model sizes, from ViT-S to ViT-B. While CroCo v2 edges out Midway Network on optical flow for ViT-B, Midway Network does not suffer the same tradeoff on semantic segmentation performance as CroCo v2. Figure 4 and Figure 5 compare predicted segmentation masks for BDD100K, and optical flow for FlyingThings and MPI-Sintel, respectively, across different methods.

**Walking Tours pretraining.** Table 2 shows results on WT-Sem and ADE20K semantic segmentation, and FlyingThings and MPI-Sintel optical flow benchmarks after WT-Venice pretraining. Again, Midway Network is the only method to achieve strong, competitive performance on *both* semantic segmentation and optical flow tasks. Note that PooDLe was pretrained at high resolution ($512 \times 1024$) and utilized external supervised optical flow networks. We include additional visualizations of predicted segmentation masks and optical flow for WT-Venice pretraining in Appendix C.

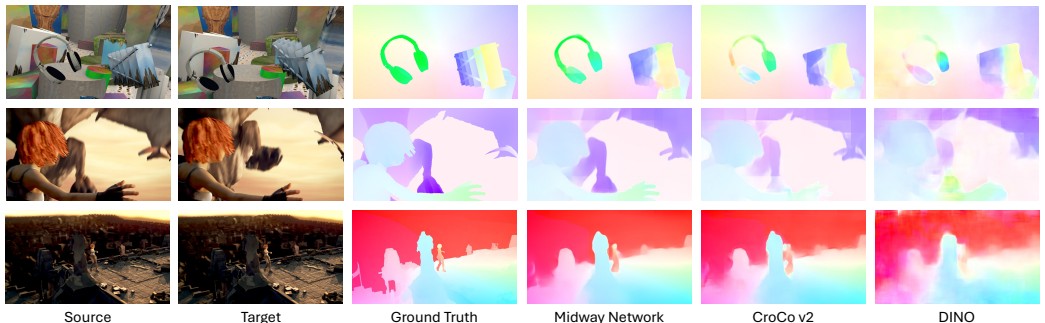

Figure 5: Visualization of FlyingThings and MPI-Sintel optical flow evaluations after finetuning. Midway Network is able to generate more accurate optical flow predictions compared to CroCo v2.

Table 3: Ablation studies on Midway Network components evaluated on BDD100K semantic segmentation linear readout and MPI-Sintel optical flow finetuning.

| Variant | Latent Dynamics | Backward | Multi-Level | Refinement | Gating | ↑mIoU | ↓EPE |
|---|---|---|---|---|---|---|---|
| 1 Base model | | | | | | 28.3 | 6.2 |
| 2 | ✓ | | | | | 30.4 | 4.4 |
| 3 | ✓ | ✓ | | | | 30.0 | 5.0 |
| 4 | ✓ | ✓ | ✓ | | | 30.4 | 5.2 |
| 5 | ✓ | ✓ | ✓ | ✓ | | 31.1 | 3.9 |
| 6 Full model | ✓ | ✓ | ✓ | ✓ | ✓ | 31.5 | 4.1 |
| 7 No backward | ✓ | | ✓ | ✓ | ✓ | 30.4 | 3.7 |
| 8 No multi-level | ✓ | ✓ | | ✓ | ✓ | 30.3 | 5.2 |
| 9 No refinement | ✓ | ✓ | ✓ | | ✓ | 30.8 | 5.1 |

Table 4: Ablation studies on Midway Network model capacity and time delta sampling hyperparameter evaluated on BDD100K semantic segmentation linear readout and MPI-Sintel optical flow finetuning.

(a) Capacity of midway path and forward dynamics

| Ablation | ↑mIoU | ↓EPE |
|---|---|---|
| Full model | 31.5 | 4.1 |
| 0.5× midway dim | 31.3 | 6.7 |
| 2× midway dim | 31.0 | 3.3 |
| 1-layer midway | 31.9 | 6.9 |
| 2-layer midway | 31.2 | 6.4 |
| 1-layer forward | 29.6 | 5.0 |
| 2-layer forward | 30.2 | 4.8 |

(b) Time delta for sampling frames

| Time Delta | ↑mIoU | ↓EPE |
|---|---|---|
| 0.16 sec | 31.0 | 4.1 |
| 0.5 sec | 32.0 | 4.3 |
| 1 sec | 31.2 | 4.6 |
| 2 sec | 31.0 | 4.9 |

## 4.3 ABLATION STUDIES

We perform a series of ablation studies, shown in Table 4, where we cumulatively add components of Midway Network until we reach the full model. For the ablations, we pretrain variants of Midway Network on BDD100K for 100 epochs and evaluate on BDD semantic segmentation with linear readout and on MPI-Sintel optical flow (clean renderings) after finetuning on FlyingChairs and FlyingThings following CroCo V2 and prior optical flow methods. For reference, we run 5 seeds for the full model (row 6), and obtain a standard deviation of 0.06 on mIoU and 0.08 on EPE. More technical details are found in Appendix B.

First, we find that adding latent dynamics modeling immediately adds a large boost to performance (row 2). Next, we observe that the hierarchical structure of the backward network and multi-level learning work together with motion latent refinement to provide further gains on both recognition and motion understanding (row 5). Finally, using gating units improves recognition (row 6) as well as visual interpretability of the learned dynamics, as shown in Figure 6. We also see that removing any of the introduced design components from Midway Network harms performance by a decent margin (rows 7 - 9).

We also investigate how varying capacity of the midway path and forward dynamics model affects performance in Table 4a. Reducing midway path capacity primarily harms EPE, whereas adding capacity improves EPE and hurts mIoU, likely because motion latents can capture more information from paired frames, but the forward prediction objective is made easier. Performance drops with fewer forward model layers. We further show in Table 4b that Midway Network is relatively robust to different time deltas for sampling frames in pretraining.

## 4.4 ANALYSIS OF DYNAMICS

To probe the extent to which Midway Network has learned dynamics after pretraining on natural videos, we introduce a new analysis method based on forwarded feature perturbation. First, we encode a pair of frames to get features $z_t$ and $z_{t+1}$ and compute motion latents $m$ between them, as usual. Then, we sample a random vector $r \sim \mathcal{N}(0, 1)$ and "perturb" a selected spatial feature by associating $r$ as a tangent vector to the selected feature in the source frame. We perform forward prediction to propagate the perturbation to the predicted target features' tangent vectors — the propagation is done

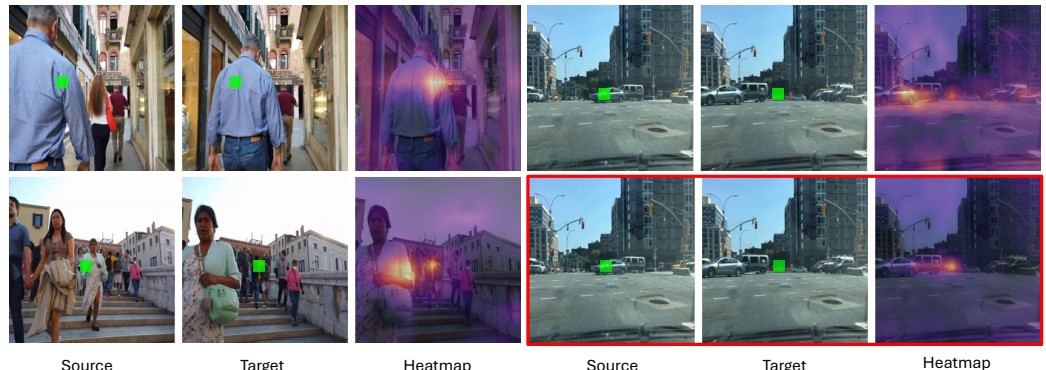

Figure 6: Heatmaps from forwarded feature perturbation. Features are perturbed at green squares in Source, which are also depicted in Target at the same location to highlight the motion between frames. Midway Network without gating units exhibits identity bias (bottom right, red border).

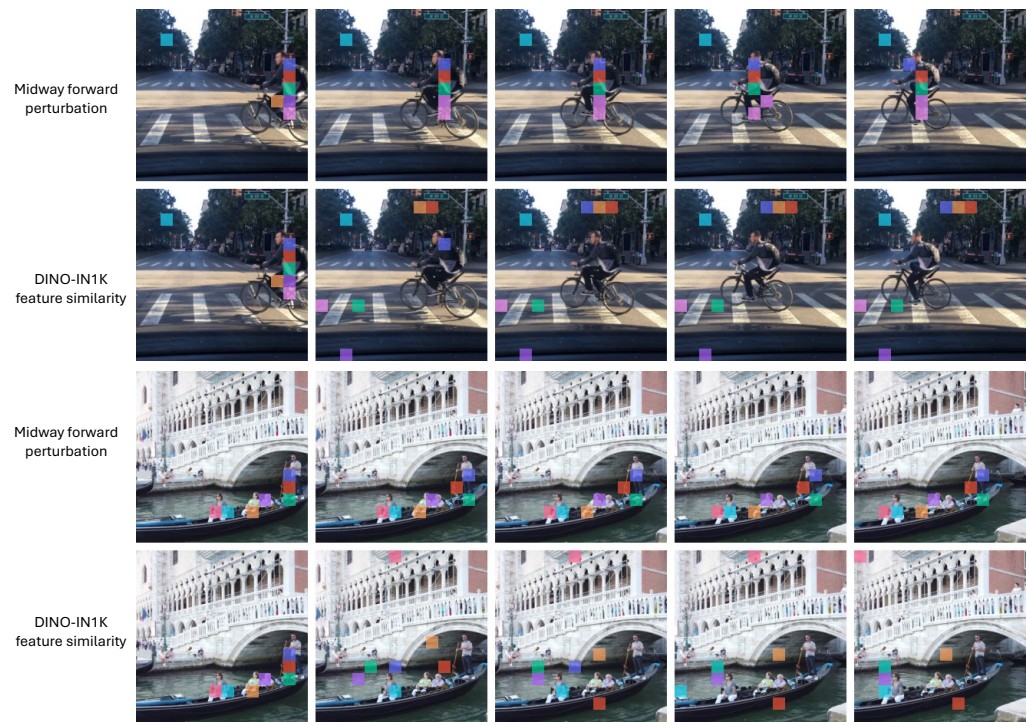

Figure 7: High-level tracking using forwarded feature perturbation and/or feature similarity. Midway Network is able to track high-level regions such as the cyclist's foot (top row, pink square).

via forward mode automatic differentiation. The cosine similarity between the random vector and the tangent vectors of the predicted features then represents the sensitivity of each spatial feature in the target frame to the initial perturbation. This process is repeated $k$ times, and the similarity scores are averaged to obtain a final heatmap over the target frame spatial locations. In Figure 6, we observe that the highest similarity regions in Target correctly correspond with the initial perturbation locations in Source (green square), indicating that the dynamics models can capture high-level correspondences. We also see that Midway Network without gating units (bottom right, red border) learns an incorrect identity mapping where the highest similarity region is the same location as the initial perturbation.

We may also use forwarded feature perturbation as a form of high-level tracking. First, for consecutive pairs of frames, we compute perturbation heatmaps over the target spatial features by individually perturbing each spatial feature in the source frame. Then, for the first frame of the video, we select an initial location and take the top-5 locations in the next frame with the highest perturbation heatmap scores; from these locations, we select the one with the highest feature similarity. This process repeats with the newly selected location until we have a track across all

frames. Figure 7 shows these tracking results in comparison to selecting the next location based on highest feature similarity with DINO (Caron et al., 2021) pretrained on ImageNet (IN1K). Despite being trained in latent space, Midway Network is able to roughly track high-level regions over time, whereas the DINO-IN1K feature similarity baseline tracks quickly diverge.

In Figure 8, we further compare the perturbation heatmaps to optical flow estimated from RAFT Teed & Deng (2020) as well as heatmaps produced by cosine similarity of last-layer dense features from different models. We convert heatmaps to optical flow by computing the (x, y) delta from each source token to the target token with highest heatmap value. We show the highest-valued target token, "Pred (K=1)," and token with highest feature similarity out of the top-2 highest-valued tokens, "Pred (K=2)." To get optical flow at the token level, "GT (Token)," we sample optical flow at the center of each source token and retrieve the mapped to target token. Forwarded feature perturbation is less noisy and more well-aligned with RAFT's optical flow compared to feature similarity baselines.

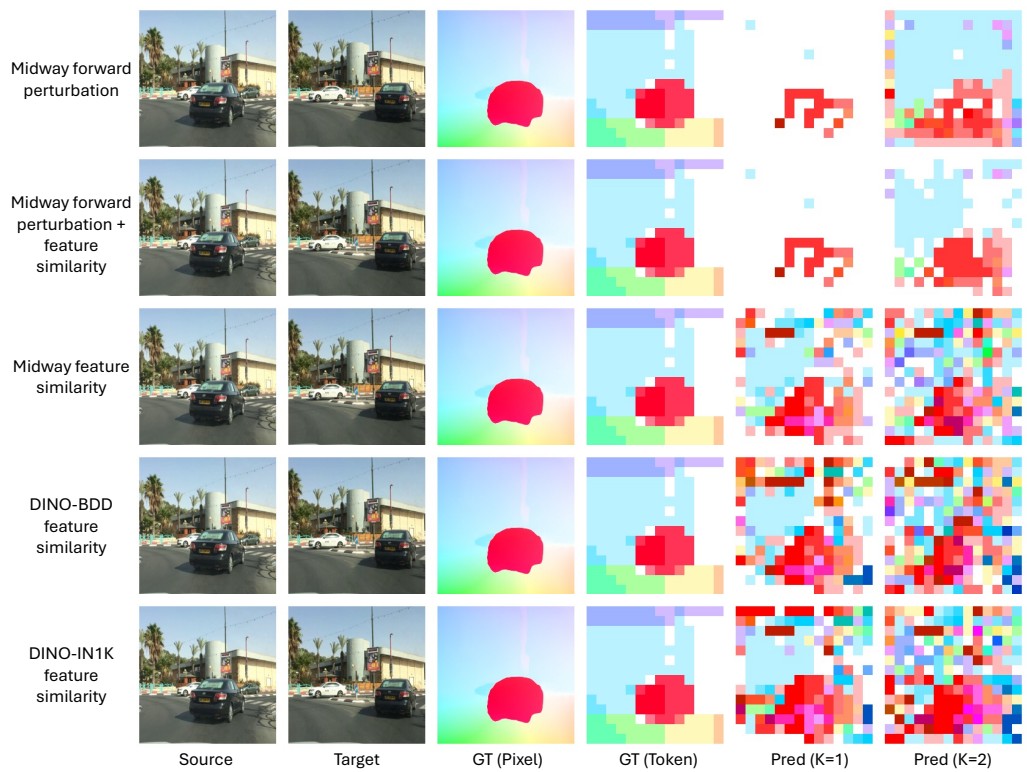

Figure 8: Optical flow estimates (Pred) derived from forwarded feature perturbation and feature similarity heatmaps compared to RAFT-predicted optical flow maps (GT) on BDD.

## 5 CONCLUSION

Object recognition and motion understanding are complementary aspects of perception, yet most self-supervised methods have focused on learning representations for only one facet. We aim to bridge this gap by extending latent dynamics modeling to the natural video domain. In this work, we propose Midway Network, the first self-supervised learning architecture to learn representations for both recognition and motion solely from natural videos, leveraging an inverse dynamics midway path, a dense forward prediction objective, and a hierarchical structure to capture the complex, multi-object scenes. Midway Network learns strong image-level representations for both recognition and motion, and in many cases, outperforms prior approaches on semantic segmentation and optical flow estimation. We have demonstrated that Midway Network can be used across different video datasets and scales well with larger models — training on more diverse data and continuing to scale model capacity could further improve performance. An exciting avenue for future work is to leverage the motion and dynamics captured by Midway Network for real-world planning tasks. Possible next steps towards this direction include incorporating action-labeled data and using Midway Network's forward dynamics predictor within a world modeling framework.

ACKNOWLEDGEMENT

We thank members of the NYU Agentic Learning AI Lab for their helpful discussions. CH is supported by the DoD NDSEG Fellowship. The work is supported in part by the Institute of Information & Communications Technology Planning Evaluation (IITP) under grant RS-2024-00469482, funded by the Ministry of Science and ICT (MSIT) of the Republic of Korea in connection with the Global AI Frontier Lab International Collaborative Research. The compute is supported by the NYU High Performance Computing resources, services, and staff expertise.

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

APPENDIX

# A ADDITIONAL RESULTS

## A.1 LONGER PRETRAINING

We provide additional experiments on WT-Venice pretraining below. Table 5 shows that Midway Network's downstream performance continues to improve with longer pretraining.

Table 5: Semantic segmentation and optical flow evaluations for additional experiments on WT-Venice $224 \times 224$ resolution pretraining. Sem. Seg. and optical flow are conducted with finetuning.

| Method | Arch | Ep. | WT-Sem Sem. Seg. UperNet | | ADE20K Sem. Seg. UperNet | | Optical Flow | | | |
| | | | | | | | FlyingThings | | MPI-Sintel | |
| | | | ↑mIoU | ↑Acc | ↑mIoU | ↑Acc | ↓EPE (c) | ↓EPE (f) | ↓EPE (c) | ↓EPE (f) |
|---|---|---|---|---|---|---|---|---|---|---|
| Midway | ViT-S | 100 | 13.1 | 85.4 | 33.4 | 76.9 | 7.7 | 7.4 | 5.2 | 6.6 |
| Midway | ViT-S | 300 | 14.8 | 86.5 | 36.9 | 78.2 | 7.3 | 6.9 | 4.0 | 5.1 |

## A.2 ADE20K LINEAR READOUT

Table 6 shows evaluation results for ADE20K semantic segmentation linear readout. Performance trends follow the UperNet finetuning results in Table 2. Again, Midway Network is competitive with baselines, PooDLe and DoRA, and furthermore, it does not rely on an external supervised optical flow network and can jointly learn representations for motion understanding.

Table 6: ADE20K semantic segmentation linear readout evaluations for WT-Venice $224 \times 224$ resolution pretraining. [†]PooDLe on $512 \times 1024$ resolution pretraining from their original table (Wang et al., 2025).

| Method | Arch | Ep. | ↑mIoU | ↑Acc |
|---|---|---|---|---|
| PooDLe[†] (Wang et al., 2025) | R50 | 20 | **14.6** | 59.0 |
| MAE (He et al., 2022) | ViT-S | 100 | 7.4 | 55.1 |
| VideoMAE (Tong et al., 2022) | ViT-S | 100 | 0.8 | 28.6 |
| DINO (Caron et al., 2021) | ViT-S | 100 | 6.9 | 48.2 |
| CroCo v2 (Weinzaepfel et al., 2022) | ViT-S | 100 | 4.2 | 48.7 |
| DoRA (Venkataramanan et al., 2024) | ViT-S | 100 | 14.1 | **63.5** |
| Midway | ViT-S | 100 | 12.1 | 61.3 |

## A.3 OPTICAL FLOW FROZEN READOUT

Table 7 provides evaluation results for optical flow linear readout. Here, the backbone parameters of each method are frozen and only the DPT (Ranftl et al., 2021) head is trained using the same data as the optical flow finetuning experiments. Midway Network's learned representations again achieve strong performance relative to the baselines.

Table 7: Optical flow frozen readout evaluations for BDD100K $224 \times 224$ resolution pretraining.

| Method | Arch | Ep. | FlyingThings | | MPI-Sintel | |
| | | | ↓EPE (c) | ↓EPE (f) | ↓EPE (c) | ↓EPE (f) |
|---|---|---|---|---|---|---|
| iBOT (Zhou et al., 2021) | ViT-S | 800 | 20.5 | 20.3 | 13.9 | 14.6 |
| DINO (Caron et al., 2021) | ViT-S | 300 | 19.0 | 17.5 | 14.0 | 13.5 |
| VideoMAE (Tong et al., 2022) | ViT-S | 300 | 20.0 | 20.0 | **11.6** | 12.2 |
| CroCo v2 (Weinzaepfel et al., 2023) | ViT-S | 300 | 39.0 | 39.2 | 24.0 | 23.9 |
| DoRA (Venkataramanan et al., 2024) | ViT-S | 300 | 20.7 | 20.6 | 12.6 | 13.3 |
| Midway (enc. only) | ViT-S | 300 | **18.8** | **17.0** | 12.5 | **11.7** |
| Midway | ViT-S | 300 | 20.2 | 19.3 | 12.8 | 12.6 |
| DINO (Caron et al., 2021) | ViT-B | 300 | **19.0** | **17.4** | 14.2 | 13.2 |
| CroCo v2 (Weinzaepfel et al., 2023) | ViT-B | 300 | 39.2 | 39.2 | 24.0 | 24.1 |
| Midway | ViT-B | 300 | 21.7 | 20.2 | **13.7** | **12.9** |

# B IMPLEMENTATION DETAILS

In this section, we provide additional details on the implementation of Midway Network, the pretraining and evaluation setups, and compute resources used for our experiments. The experiments were implemented using the `PyTorch` framework.

### B.1 ARCHITECTURE

The ViT encoders have 12 feature levels, and we perform the dense forward prediction objective at levels 3, 6, and 9. The midway path infers motion latents with feature inputs at level 12 for the level 9 objective and refines them as described in Section 2 for levels 6 and 3. The midway inverse dynamics model at each level is a 4-block transformer with feature dimension of 192, with linear projectors to map from and to the original feature dimension. We use 10 learnable tokens for the motion latents. The backward layers are 1-block cross-attention transformers with feature dimension equal to the dimension of the underlying ViT encoder, i.e. 384 for ViT-S and 768 for ViT-B. The forward dynamics model at each level is a 4-block transformer with feature dimension equal to the underlying encoder dimension as well. The learnable gating units are placed at all but the first block. Each gating unit is a multi-layer perceptron with 1 hidden layer of same dimension as the encoder, GELU activation, and a final sigmoid activation. To bias the initial gating weights towards 1, i.e. the original fully-weighted residual connection, we add a bias of 4 to the input of the sigmoid.

We follow DINO (Caron et al., 2021) for implementation of the joint-embedding invariance objective, using the same projection heads, centering and sharpening operations, and temperature schedules as described in their paper. Given that we have 2 paired video frames as input, we can sample 2 global crops and 8 local crops from each frame and compute the loss between crops across frames to leverage the natural temporal motion augmentation. The loss is also symmetrical, where we compute the loss for the original frame ordering as well as the reversed ordering. We utilize this setup for the DINO baseline as well for fair comparison. The final loss is an equal-weighted sum of the dense forward prediction loss, averaged over the feature levels, and the joint-embedding invariance loss:

$$\mathcal{L} = \frac{1}{L} \sum_{l=1}^{L} \mathcal{L}_{dyn}^{l} + \mathcal{L}_{inv}. \tag{3}$$

### B.2 PRETRAINING

We outline the hyperparameters used for pretraining in Table 8. The hyperparameters largely follow the DINO (Caron et al., 2021) training recipe. We use the same hyperparameters for BDD100K and Walking Tours pretraining. For BDD100K, we utilize repeat sampling following MAE-st (Feichtenhofer et al., 2022), which samples $R = 5$ frames each time a video is seen for faster data loading. Therefore, we treat each pass through the dataset as $R$ epochs.

Table 8: Hyperparameters used for full Midway Network experiments.

| Hyperparameter | Value |
| --- | --- |
| Learning rate | $5 \times 10^{-4}$ |
| Learning rate warmup | 10 epochs |
| Learning rate schedule | cosine |
| Batch size | 200 |
| Weight decay | 0.04 |
| Weight decay end | 0.4 |
| Optimizer | AdamW |
| Betas | (0.9, 0.999) |
| Gradient clip norm | 3.0 |
| Drop path rate | 0.1 |
| Use FP16 | True |

### B.3 BASELINES

We use the official implementations to pretrain the baselines on BDD100K and Walking Tours. We use the released checkpoints for DINO, DoRA, and PooDLe on Walking Tours; semantic segmentation finetuning results for MAE, DINO, DoRA, and PooDLe are also from the original table in PooDLe (Wang et al., 2025).

### B.4 EVALUATION

For the semantic segmentation tasks, we follow the ViT-based setup described in PooDLe (Wang et al., 2025), based on the `mmsegmentation` (Contributors, 2020) codebase. The linear and UperNet readout setups for BDD100K and CityScapes were originally from FlowE (Xiong et al., 2021); the UperNet finetuning setup for ADE20K was originally from iBOT (Zhou et al., 2021).

For the optical flow tasks, we follow the finetuning evaluation setup described in CroCo v2 (Weinza-epfel et al., 2023) and use their official implementation. Our main results follow CroCo v2's setup for Table 1 from their paper; our ablation studies follow their setup for their Table 11 ("smaller training data") to match the settings of other optical flow methods. The primary difference is that we replace CroCo v2's decoder with Midway Network's midway inverse dynamics and forward dynamics models. We use the following as input to the DPT (Ranftl et al., 2021) that outputs the optical flow predictions: dense tokens of encoder feature level 12, dense spatial tokens corresponding to the target frame processed by the midway model at the highest level of the dense objective, dense token prediction of the forward model at the highest objective level, and dense token prediction of the forward model at the lowest objective level. For reference, the midway model processes the dense spatial tokens from the source and target frames alongside the motion latents. We use this architecture for all other baselines besides CroCo v2 with randomly initialized weights, as they do not have binocular components.

## B.5 COMPUTE AND TRAINING COSTS

Table 9 provides a comparison on training cost in FLOPs per single training example and model size in parameters for Midway Network and the baseline methods. Midway Network uses less than half of the FLOPs of prior video data-based learning methods, PooDLe and DoRA. The dynamics networks of Midway Network use more parameters to capture motion information, but avoid costly iterative refinement operations used by prior flow methods such as RAFT (Teed & Deng, 2020) and FlowFormer (Huang et al., 2022). Table 10 shows the compute resources used for the experiments.

Table 9: Training cost (GLOPs per example) and model size (millions of parameters) of Midway Network and baseline methods.

| Method | Training cost (GFLOPs) | Parameters (millions) |
|---|---|---|
| Midway Network | 90.8 | 21.7 (encoder), 36.6 (dynamics networks) |
| PooDLe | 202.3 | 23.5 (encoder), 12.1 (spatial decoder) |
| DoRA | 202.1 | 21.7 (encoder) |
| CroCo v2 | 6.9 | 21.7 (encoder), 7.2 (decoder) |
| DynaMo | 68.9 | 21.7 (encoder), 13.0 (dynamics networks) |
| VideoMAE | 11.6 | 22.0 (encoder), 2.0 (decoder) |
| iBOT | 35.3 | 21.7 (encoder) |
| DINO | 50.4 | 21.7 (encoder) |

Table 10: Compute resources and time used for Midway Network experiments.

| Experiment | Epochs | Resources | Time |
|---|---|---|---|
| BDD100K ViT-S pretraining | 300 | 2 A100 GPUs | 66 hours |
| BDD100K ViT-B pretraining | 300 | 8 RTX A6000 GPUs | 27 hours |
| BDD100K ViT-S ablations | 100 | 2 A100 GPUs | 24 hours |
| Walking Tours ViT-S pretraining | 100 | 4 RTX A6000 GPUs | 29 hours |

## C   MORE VISUALIZATIONS

We show additional visualizations of predictions from the semantic segmentation evaluations in Figure 9 for CityScapes, Figure 9 for WT-Sem, and Figure 11 for ADE20K, and optical flow evaluations for models pretrained on Walking Tours in Figure 12. We also provide visualizations of optical flow evaluations comparing Midway Network and CroCo v2 for different model sizes in Figure 13.

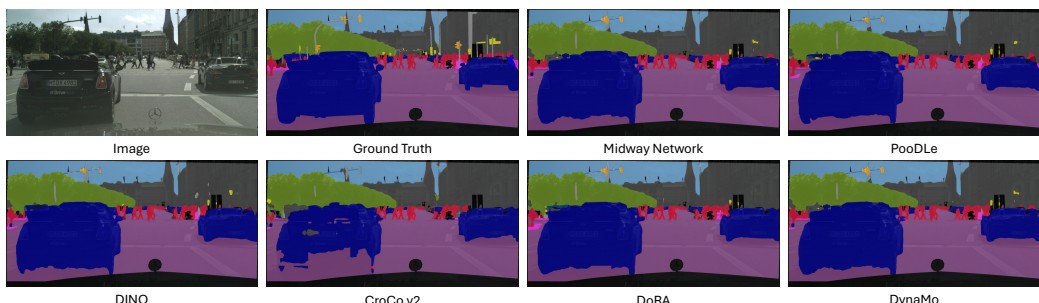

Figure 9: Visualization of CityScapes semantic segmentation UperNet readout. Midway Network generates cleaner boundaries, particularly for the crossing pedestrians.

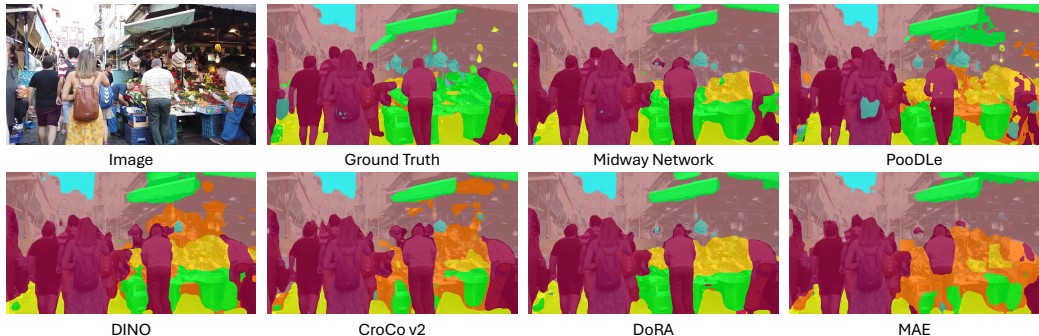

Figure 10: Visualization of WT-Sem semantic segmentation UperNet finetuning. Midway Network is able to produce reasonable segmentation masks, even in cluttered scenes.

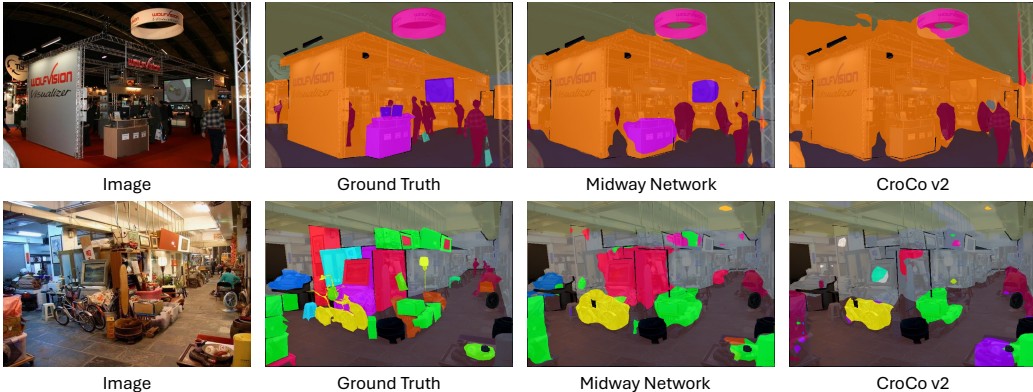

Figure 11: Visualization of ADE20K semantic segmentation UperNet finetuning. Midway Network generates more accurate segmentation masks compared to CroCo v2.

We also include more examples of the forwarded feature perturbation analysis of Midway Network's learned dynamics, with heatmaps in Figure 14 and high-level tracking in Figure 15.

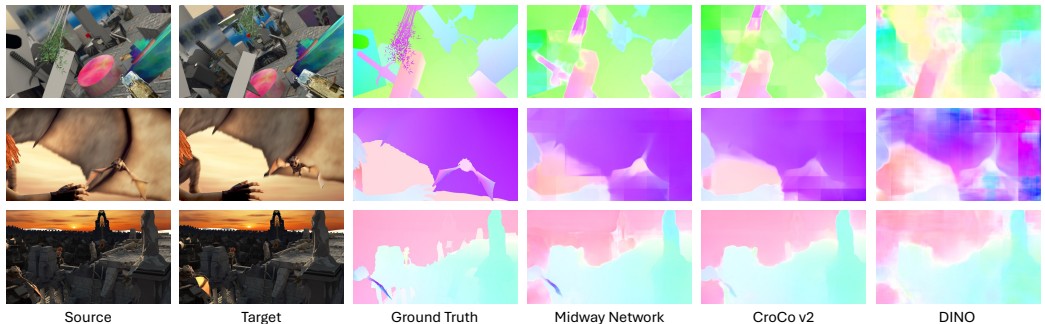

Figure 12: Visualization of FlyingThings and MPI-Sintel optical flow evaluations after finetuning for models pretrained on WT-Venice. Midway Network is able to generate more accurate optical flow predictions compared to CroCo v2 and DINO.

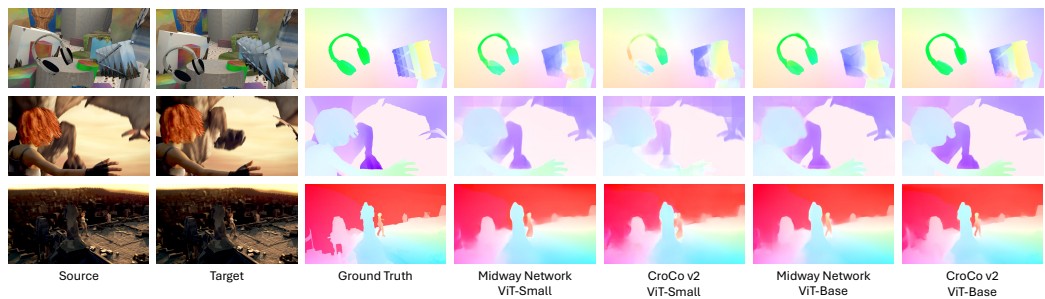

Figure 13: Visualization of FlyingThings and MPI-Sintel optical flow evaluations after finetuning for Midway Network and CroCo v2 pretrained on BDD for varying model sizes. Moving from ViT-Small to ViT-Base primarily provides fine-grained improvements in optical flow estimation.

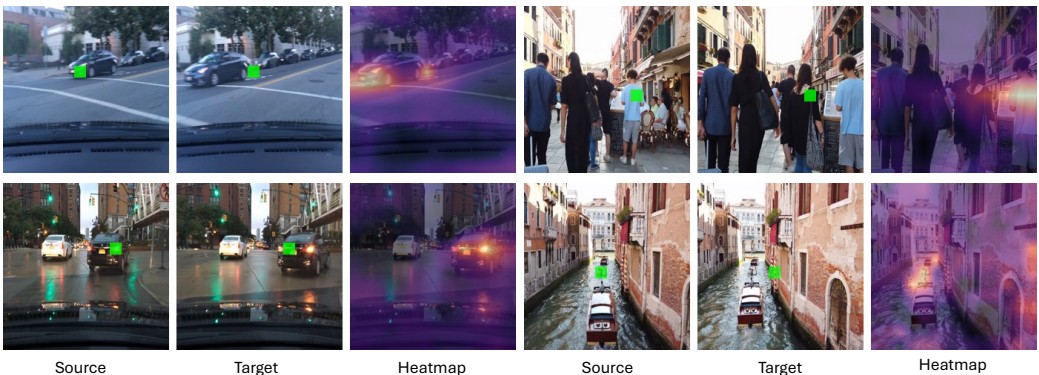

Figure 14: Heatmaps for forwarded feature perturbation in Source (green squares); shown in Target at the same location to highlight motion. The learned dynamics can capture high-level correspondence, such as the right taillight of the black car (bottom left).

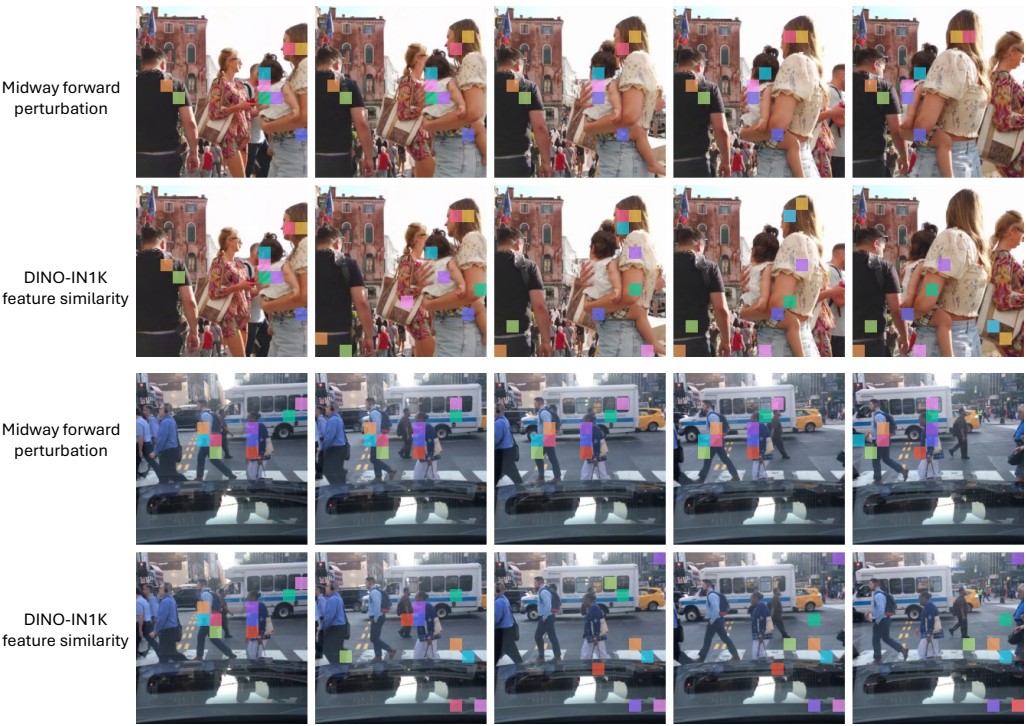

Figure 15: High-level tracking using forwarded feature perturbation and/or feature similarity. Midway Network is able to track high-level regions through motion transformations, such as the back of the toddler (top row, pink square).

