# OpenReview forum: "Midway Network: Learning Representations for Recognition and Motion from Latent Dynamics"
_ICLR.cc/2026/Conference — ICLR 2026 Poster_

### Official Review · Reviewer_m9iU · 2025-10-18

**Soundness:** 3
**Presentation:** 3
**Contribution:** 3
**Rating:** 6
**Confidence:** 4

**Summary:**

The authors, in this work, introduce a new self-supervised learning (SSL) architecture called Midway Network.

- They extend the concept of latent dynamics modeling, used in control and decision-making, to the visual perception domain.

- Midway network is a self-supervised video model designed to learn both: Object recognition (semantic understanding) and Motion understanding (how things move over time) purely from natural, unlabeled videos — without relying on curated image datasets or external motion labels like optical flow.

- Midway Network outperforms or matches the best SSL baselines (like DINO, DoRA, CroCo v2) on both tasks: Semantic segmentation and Optical flow estimation.

**Strengths:**

### 1. Conceptual Soundness
- The authors ground their approach in predictive coding and latent dynamics modeling, both well-established ideas in neuroscience and machine learning.

- The theoretical motivation, that perception arises from predicting sensory changes is sound.

- The authors identify a clear gap: self-supervised models typically learn either recognition (DINO, iBOT) or motion (CroCo, FlowE), not both.

### 2. Architectural Novelty
- The architecture is a careful combination of: 1. Inverse dynamics (midway path) to infer motion, 2. Forward prediction (dense feature-level) to learn temporal coherence, and 3. Hierarchical refinement (inspired by optical flow networks e.g., PWC-Net, RAFT).

- The design is justified by analogies to biological systems and prior SSL hierarchies (e.g., Ladder Networks, DINO).

- The inclusion of gating units in transformer residuals is thoughtful as it prevents trivial identity mappings, a known issue in predictive models.

### 3. Experimental Soundness
- Comprehensive evaluations across recognition (semantic segmentation) and motion (optical flow).

- Extensive comparison against relevant baselines: DINO, DoRA, PooDLe, CroCo v2, DynaMo, etc.

- Consistent use of natural video datasets (BDD100K and Walking Tours) supports the claim of "learning from natural data only."

- Sufficient ablation studies test each architectural component's impact on both semantic and motion metrics.

### 4. Interpretation Soundness
- The "forwarded feature perturbation" method is a novel and interpretable way to visualize what the dynamics model learns.

- It qualitatively demonstrates non-trivial motion correspondence — a rare strength in SSL papers.

**Weaknesses:**

### 1. Conceptual Limitations
- While conceptually coherent, "latent dynamics" is borrowed from control/planning literature and adapted here somewhat heuristically — the paper lacks a strong theoretical derivation connecting latent dynamics to semantic learning in videos.

- The link between motion prediction and semantic invariance is intuitive but not formally analyzed.

### 2. Architectural Limitations
- The midway and backward paths add significant complexity; it’s unclear if all components are essential (though the ablation studies help).

- The architecture might overfit to short temporal correlations (1-second gaps between frames) and may not generalize to longer-horizon motion.

### 3. Experimental Limitations
- Only two pretraining sources (BDD and WT-Venice). That’s small compared to large-scale pretraining regimes (e.g., Kinetics-700, Ego4D). Therefore, it is hard to tell if results scale to diverse or indoor/outdoor mixed environments.

- The authors do not measure performance over multiple time steps (e.g., predicting motion 10 frames ahead).

- Some of the baselines (e.g., PooDLe) use higher resolution or external flow networks, making cross-comparison imperfect.

- Few metrics report standard deviation or multiple seeds.

### 4. Interpretation Limitations:
- The forward feature perturbation analysis is qualitative. There’s no quantitative measure of how well it aligns with ground-truth motion.

- The theoretical justification for why this analysis reflects "high-level correspondence" is intuitive but not formalized.

**Questions:**

Please see the discussion in weakness section.

---

> ### Author Response · Authors · 2025-11-21
> **Response to Reviewer m9iU [1/2]**
>
> Thank you for taking the time to review our work and provide helpful feedback. We appreciate that you found our paper identifies a clear gap in prior self-supervised learning works, the method design to be justified and thoughtful, the evaluations to be comprehensive and extensive, and the forward feature perturbation method to be novel and interpretable. We would like to address your comments below.
>
> > ### While conceptually coherent, "latent dynamics" is borrowed from control/planning literature and adapted here somewhat heuristically — the paper lacks a strong theoretical derivation connecting latent dynamics to semantic learning in videos.
>
> To clarify the theoretical link between latent dynamics and semantic learning, we start from how perception, which includes semantic learning, arises from predicting sensory changes, i.e. predictive coding, which we discuss in Section 2 following several theoretical works like Friston’s theory of cortical responses [1]. Latent dynamics itself is a form of predictive coding, where we are predicting the latent future representations conditioned on inferred motion latents. Therefore, leveraging latent dynamics is directly connected to perception and consequently, semantic learning, following predictive coding theory.
>
> > ### The link between motion prediction and semantic invariance is intuitive but not formally analyzed.
>
> In the ablation studies (Table 3), we demonstrate that by adding latent dynamics, i.e. motion prediction related components, semantic segmentation performance improves, indicating that the model has improved semantically invariant features.
>
> > ### The midway and backward paths add significant complexity; it’s unclear if all components are essential (though the ablation studies help).
>
> We believe the ablation studies in Table 3 show that each component has a notable impact in performance. For instance, removing either the backward network, multi-level learning, or refinement design for the midway path results in a significant drop in mIoU, indicating the contribution of each of these components.
>
> > ### The architecture might overfit to short temporal correlations (1-second gaps between frames) and may not generalize to longer-horizon motion.
>
> We note that motion in videos is not consistent, so a 1-second gap can correspond to different motion patterns, e.g., a car waiting at a stop light versus driving on a highway. However, we agree that the time used to sample frames is a hyperparameter; we show below in Table A that Midway Network is relatively robust to training with different time deltas. This suggests we can train with larger ranges of time deltas to reduce overfitting. We have also included this ablation table in **Appendix A.3, Table 6** of the revised paper (green text).
>
> | Time gap between frames | mIoU | EPE |
> | :--- | :--- | :--- |
> | 0.16 seconds | 31.0 | 4.1 |
> | 0.5 seconds | 32.0 | 4.3 |
> | 1.5 seconds | 31.2 | 4.6 |
> | 2 seconds | 31.0 | 4.9 |
>
> Table A. Semantic segmentation and optical flow evaluations for Midway Network pretrained on BDD using different time gaps between paired frames.
>
> > ### Only two pretraining sources (BDD and WT-Venice). That’s small compared to large-scale pretraining regimes (e.g., Kinetics-700, Ego4D). Therefore, it is hard to tell if results scale to diverse or indoor/outdoor mixed environments.
>
> We would like to highlight that BDD is within a similar order of magnitude as Kinetics-700 and Ego4D, with 778 hours of video compared to 1330 hours for Kinetics-700 and 3670 hours for Ego4D. Furthermore, BDD and WT contain a diverse range of objects and scenes across different cities as well as weather, time, and lighting conditions. We also evaluate our models on ADE20K, which contains numerous indoor scenes such as bedrooms, kitchens, and offices. However, overall, we agree with your suggestion that exploring scaling to more datasets is a good step for future work.
>
> > ### The authors do not measure performance over multiple time steps (e.g., predicting motion 10 frames ahead).
>
> Please see Table A in our response above, where we show that Midway Network performs well when trained with different time deltas.
>
> > ### Some of the baselines (e.g., PooDLe) use higher resolution or external flow networks, making cross-comparison imperfect.
>
> This gives an advantage to baseline methods like PooDLe, which are given more information per frame due to the higher resolution and use external optical flow networks which were trained with ground-truth supervision. In particular, Midway Network improves over PooDLe by removing the need for an external flow network.
>
> > ### Few metrics report standard deviation or multiple seeds.
>
> We report standard deviation over multiple seeds for the ablation studies in Section 4.3. Due to computational constraints, we were unable to run multiple seeds for the larger main tables.
>
> [1] Friston, Karl. “A theory of cortical responses.” Philosophical transactions of the Royal Society of London. 2005

---

> > ### Author Response · Authors · 2025-11-21
> > **Response to Reviewer m9iU [2/2]**
> >
> > > ### The forward feature perturbation analysis is qualitative. There’s no quantitative measure of how well it aligns with ground-truth motion.
> >
> > We note that while the forwarded feature perturbation analysis is qualitative, the optical flow results in Table 1 and Table 2 serve as a quantitative measure of how well the Midway Network representations align with ground-truth motion. However, it is possible to also evaluate the heatmaps produced by forwarded feature perturbation using optical flow maps. First, as forwarded feature perturbation is on the token level whereas optical flow maps are on the pixel level, we sample the optical flow at the center of each token in the source frame and retrieve the target token that the optical flow maps to in the target frame (we also tried the optical flow averaged across the token with similar results). With this token-level optical flow, we compute the top-K accuracy of the forwarded feature perturbation heatmaps, i.e., for a source token, is the target token linked via optical flow within any of the top-K most similar target tokens in the heatmap? We use RAFT, a supervised-trained optical flow model, to compute optical flow maps for BDD. We then compare to heatmaps computed from cosine similarity of the last-layer dense features between the source and target frames of various baseline models. Table A below shows these quantitative results, where forwarded feature perturbation heatmaps are more well-aligned to the optical flow maps than feature similarity heatmaps from baseline models. The results are averaged over frame pairs, each randomly sampled from a random subset of 320 videos. We also provide visualizations of these heatmaps, converted to optical flow-like maps, compared to optical flow maps generated from RAFT in **Appendix D, Figure 15** of the revised paper (green text).
> >
> > | Method | Top-1 | Top-2 | Top-3 |
> > | :--- | :--- | :--- | :--- |
> > | Midway Network - BDD: Forwarded Perturbation | 73.93% | 88.20% | 92.39% |
> > | Midway Network - BDD: Feature Similarity | 66.19% | 75.73% | 80.67% |
> > | DINO - BDD: Feature Similarity | 58.59% | 68.66% | 74.27% |
> > | DINO - ImageNet: Feature Similarity | 55.29% | 66.21% | 71.89% |
> >
> > Table A. Top-K accuracy of forwarded feature perturbation and feature similarity heatmaps evaluated against optical flow maps predicted by RAFT for BDD frame pairs.
> >
> > > ### The theoretical justification for why this analysis reflects "high-level correspondence" is intuitive but not formalized.
> >
> > We are glad you found the theoretical justification intuitive. For the formal justification, we first define “high-level correspondence” as the notion that two regions in a pair of frames should be linked due to their semantic content. Therefore, the desired analysis should capture the degree to which two regions across a pair of frames are linked and compare it with the expected correspondence mapping. The forwarded feature perturbation analysis perturbs a dense feature in the first frame $z_{ij}^{t}$ and uses the Midway Network to propagate the perturbation to the second frame dense features $z^{t+1}$. The similarity between the resulting perturbation in a particular dense feature $z_{kl}^{t+1}$ and the perturbation of the original first frame dense feature $z_{ij}^{t}$ is then simply one way to measure how the trained model links these two spatial regions. In Figure 6, we observe that these measured similarities from forwarded feature perturbation accurately match the expected semantic correspondences, e.g., the perturbation on the car in the first frame is most similar to the region containing the car in the second frame (Figure 6, top right).

---

> > > ### Comment · Reviewer_m9iU · 2025-11-25
> > >
> > > I appreciate the authors' detailed responses and the additional experiments they conducted. Several clarifications are helpful, but a number of my original concerns remain only partially addressed.
> > >
> > > **1. Theoretical link between latent dynamics and semantic learning.**
> > >
> > > The response cites predictive coding theory, but the connection between the specific training objective (dense forward prediction + DINO loss) and the emergence of semantic invariance lacks formal explanation. Section 2 mainly surveys related work rather than providing a derivation.
> > >
> > > **2. Motion prediction leads to semantic invariance.**
> > >
> > > Pointing to improvements in Table 3 is useful, but I was asking for conceptual or analytical justification, not only empirical trends.
> > >
> > > **3. Architectural complexity and component necessity.**
> > >
> > > Ablations do show that components often help, but some effects are mixed (e.g., the backward network slightly improves EPE when removed). A more nuanced interpretation would strengthen the argument.
> > >
> > > **4. Short vs. long temporal correlations.**
> > >
> > > The time-gap experiments are appreciated, but they do not address multi-step rollout or long-horizon dynamics. The architecture, as written (Algorithm 1), predicts only a single step ahead.
> > >
> > > **5. Limited pretraining diversity.**
> > >
> > > I acknowledge BDD's scale, but it remains domain-specific, and WT pretraining uses only one video. This was the basis of my concern, not dataset size alone.
> > >
> > > **6. Multi-step prediction.**
> > >
> > > The response does not directly address the question. Changing time gaps is not equivalent to evaluating multi-step prediction capability.
> > >
> > > **7. Missing standard deviations in main tables.**
> > >
> > > I understand the computational constraints, but variance in primary results would increase reliability.
> > >
> > > Overall, while the responses clarify some points, several core issues remain unresolved. Therefore, at this point, I would like to maintain my score.

---

> ### Author Response · Authors · 2025-12-03
> **Response to Reviewer m9iU**
>
> We thank you for your response, are glad the clarifications were helpful, and wish to address your remaining comments below:
>
> > ### 1. Theoretical link between latent dynamics and semantic learning.
>
> We offer the following explanation for further clarification. The dense forward prediction objective forces the prediction, conditioned on source frame features $z_t$ and latent motion $m$, to match the target frame features $z_{t+1}$. Given that the predictor has finite capacity, it is easier for it to learn a transformation function on stable semantic representations of objects than to directly model pixel-level dynamics. Thus, minimizing the prediction error encourages the encoder to learn predictable semantic representations. As noted in LAPO [1], an efficient encoding of $m$ is to represent transformations between frames, such as affine changes and deformations, allowing the encoder to still maintain equivariant features for motion, i.e., changing position and pose. The DINO objective enforces that features from the same region, which likely contains the same semantic object, to be the same under view augmentations such as spatial jittering and temporal motion, further encouraging semantic invariance [2, 3]. We also note that the dense forward prediction objective is an implementation of predictive coding, which has been previously linked to semantic learning.
>
> > ### 2. Motion prediction leads to semantic invariance.
>
> We provide the following conceptual explanation. Given a model that has learned motion prediction, it has also learned to link the same object in different frames together. It is more efficient for the predictor to utilize this motion information to transport features from the source frame to the target frame. This implies that the feature representation of an object should remain constant or similar despite the change across frames. Consequently, representations for the same semantic entity become invariant to these frame transformations, i.e., semantic invariance.
>
> > ### 3. Architectural complexity and component necessity.
>
> Yes, we can provide a more detailed explanation of the mixed effects. The backward network serves to incorporate high-level features into lower-level, high-frequency features to produce final feature maps at the lower level. Therefore, removing the backward network makes it slightly easier to only use lower-level, high-frequency features to tackle the fine-grained optical flow task (better EPE), but harder to obtain feature maps with both semantic and high-frequency information for semantic segmentation (worse mIoU). The backward network is not effective without multi-level and refinement because there are no lower-level, high-frequency features to mix in and no mechanism to iteratively correct motion latents with these features respectively.
>
> > ### 4. Short vs. long temporal correlations.
>
> We wish to emphasize that the focus of our paper is to propose an SSL architecture that jointly learns representations for recognition and motion understanding from natural videos, rather than to perform multi-step rollout. We identify in our conclusion (Section 5) that extending Midway Network for rollout and world modeling tasks is an exciting _future_ direction. Our experiments demonstrate that the learned latent motion contains useful motion information, and a potential next step could be to learn mappings to action / language control signals for sampling potential future rollouts.
>
> > ### 5. Limited pretraining diversity.
>
> We advocate that while BDD represents a specific domain, the experiments show that our method can work across two distinct data settings, driving scenes and human walking videos, with **identical** hyperparameters. We also note that DoRA, the original paper for Walking Tours, also primarily used WT-Venice for its main experiments. Nevertheless, we look forward to expanding to more data domains as a next step.
>
> > ### 6. Multi-step prediction.
>
> Please see response #4 above.
>
> > ### 7. Missing standard deviations in main tables.
>
> We advocate that this resource constraint is widely recognized in the field, as many recent vision foundation model papers, e.g., CLIP, DINO v1-v3, BYOL, MAE, VideoMAE, I-JEPA, V-JEPA, MoCo, CroCo v2, DoRA, iBOT, only report single-number performance.
>
> References
>
> [1] Learning to Act without Actions. Schmidt and Jiang. ICLR 2024.
>
> [2] Emerging Properties in Self-Supervised Vision Transformers. Caron et al. ICCV 2021.
>
> [3] Self-Supervised Learning from Images with a Joint-Embedding Predictive Architecture. Assran et al. ICCV 2023.

---

### Official Review · Reviewer_9dmU · 2025-10-26

**Soundness:** 2
**Presentation:** 2
**Contribution:** 2
**Rating:** 4
**Confidence:** 3

**Summary:**

This paper introduces Midway Network, a self-supervised learning framework that jointly learns object recognition and motion understanding from natural videos through latent dynamics modeling. The key idea is to integrate inverse and forward dynamics modules within a hierarchical architecture, where motion latents describe transformations between consecutive frames and are refined top-down across feature levels. Experiments show that Midway Network achieves strong performances in both semantic segmentation and optical flow tasks.

**Strengths:**

S1. Unified SSL framework for both object recognition and motion understanding: The paper presents a coherent approach that bridges predictive coding theory with modern self-supervised Transformers, achieving representation learning for both semantics and motion within a single framework.

S2. In-depth analysis: The analysis of learned motion latents in Section 4.4 (forwarded feature perturbation) is technically sound and creative. It provides interpretable evidence that the model captures spatial correspondences.

S3. Diverse Evaluation Domains: The proposed method is evaluated on several tasks and benchmarks.

**Weaknesses:**

W1. Incremental novelty: the proposed inverse + forward dynamics framework for learning latent motions follows a well-established approach used in latent world models and video prediction literature [a, b, c]. Moreover, the hierarchical refinement design is conceptually similar to the Spatial Dynamics Module (SDM) used in PooDLe..

W2. No guarantee of motion: The paper does not theoretically ensure that the learned latent $m_t$ encodes motion rather than directly leaking target feature information ($𝑧_{𝑡+1}$). Although empirical analyses (optical flow results, Sec. 4.4 perturbation study) suggest motion-like behavior, the model could still “hack” the objective by embedding target information in $𝑚$ making the representation less interpretable.

W3. Limited quantitative analysis of motion learning:
Section 4.4 provides compelling qualitative examples, but the conclusions would be stronger with quantitative validation. For instance, comparing perturbation-based correspondence against pseudo ground-truth from an off-the-shelf tracker could more rigorously substantiate the claimed motion alignment.

W4. Marginal effect of design components: As seen in Table 3 (e.g., 2→7, 3→8, 4→9 comparisons), adding design components (e.g., backward layers, gating) only yields small incremental gains, which questions their relative contribution to the overall improvement.

W5. (Minor) Lack of planning or world-model evaluation: Despite the discussion of potential applications to planning and world modeling, no experiments demonstrate this capability, slightly limiting the perceived impact of the proposed approach.

[a] Bruce et al., “Genie: Generative Interactive Environments,” ICML, 2024.\
[b] Ye et al., “Latent Action Pretraining from Videos,” ICLR, 2025.\
[c] Gao et al., “AdaWorld: Learning Adaptable World Models with Latent Actions,” ICML, 2025.

**Questions:**

Q1. Why are optical flow results for PooDLe not reported in Tables 1–2?
Was there a fundamental limitation in adapting its architecture for flow prediction, or were these experiments omitted for other reasons?

---

> ### Author Response · Authors · 2025-11-21
> **Response to Reviewer 9dmU [1/2]**
>
> Thank you for taking the time to review our work and provide valuable feedback. We appreciate that you found our method to be coherent, our novel forwarded feature perturbation method to be technical sound and creative, and our evaluations to be diverse. We would like to address your comments below.
>
> > ### Incremental novelty: the proposed inverse + forward dynamics framework for learning latent motions follows a well-established approach used in latent world models and video prediction literature [a, b, c]. Moreover, the hierarchical refinement design is conceptually similar to the Spatial Dynamics Module (SDM) used in PooDLe.
>
> We advocate that our paper proposed framework presents two main novel contributions over existing methods that use the inverse + forward dynamics framework. First, Midway Network leverages latent dynamics to learn representations for both recognition and motion and notably, is the _first_ self-supervised learning method to learn recognition and motion solely from natural videos. In comparison, Genie [a], LAPA [b], and AdaWorld [c] do not learn representations for recognition and motion understanding, and have only demonstrated their methods on game/simulated or controlled robotic environments instead of real-world natural videos. Our architecture design also outperforms DynaMo in the natural video pretraining setting, as shown in Table 1. Second, the midway path design is distinct from prior approaches, by using higher-level forward predictions to refine lower-level motion latents. In our ablations in Section 4.3, we demonstrate that this novel multi-level refinement design results in a meaningful improvement on downstream evaluations.
>
> > ### No guarantee of motion: The paper does not theoretically ensure that the learned latent $m_t$ encodes motion rather than directly leaking target feature information ($z_{t+1}$). Although empirical analyses (optical flow results, Sec. 4.4 perturbation study) suggest motion-like behavior, the model could still “hack” the objective by embedding target information in making the representation less interpretable.
>
> We note that the learned latent $m_t$ is an information bottleneck as the capacity of the latent motion is much smaller than the target features $z_{t+1}$, i.e., 10 tokens x 192 dimension << 196 tokens x 384 dimension. Thus, the latent is encouraged to compress useful information from $z_t$ and $z_{t+1}$ to satisfy the forward prediction objective, such as motion or motion-like representations. Empirical evaluations, such as the ones we show in Section 4.2 and Section 4.4, are currently the only way to demonstrate that the learned latents have captured motion information similar to how prior methods like Genie [a], LAPA [b], AdaWorld [c] use generation and manipulation experiments to empirically show that the learned latents contain action information.
>
> > ### Limited quantitative analysis of motion learning: Section 4.4 provides compelling qualitative examples, but the conclusions would be stronger with quantitative validation.
>
> We appreciate that you found the qualitative analysis compelling! Thank you for the suggestion; we have implemented a quantitative metric using optical flow maps from RAFT, a supervised-trained off-the-shelf tracker. First, because forwarded perturbation is on the token-level whereas optical flow maps are on the pixel-level, we sample the optical flow at the center of each token in the source frame and retrieve the target token that the optical flow maps to in the target frame (averaging across token has similar results). With this token-level optical flow, we compute the top-K accuracy of the forwarded perturbation heatmaps, i.e., for a source token, is the target token linked via optical flow within any of the top-K most similar target tokens in the heatmap? We use RAFT to compute optical flow maps for BDD. We then compare to heatmaps computed from cosine similarity of the last-layer dense features between the source and target frames of various baseline models. Table A below shows these results, where forwarded perturbation heatmaps are more well-aligned to the optical flow maps than feature similarity heatmaps from baseline models. The results are averaged over frame pairs, each randomly sampled from a random subset of 320 videos. We also visualize these heatmaps, converted to optical flow-like maps, compared to optical flow maps generated from RAFT in **Appendix D, Figure 15** of the revised paper (green text).
>
> | Method | Top-1 | Top-2 | Top-3 |
> | :--- | :--- | :--- | :--- |
> | Midway Network - BDD: Forwarded Perturbation | 73.93% | 88.20% | 92.39% |
> | Midway Network - BDD: Feature Similarity | 66.19% | 75.73% | 80.67% |
> | DINO - BDD: Feature Similarity | 58.59% | 68.66% | 74.27% |
> | DINO - ImageNet: Feature Similarity | 55.29% | 66.21% | 71.89% |
>
> Table A. Top-K accuracy of forwarded perturbation and feature similarity heatmaps evaluated against optical flow maps predicted by RAFT for BDD frame pairs.

---

> > ### Author Response · Authors · 2025-11-21
> > **Response to Reviewer 9dmU [2/2]**
> >
> > > ### Marginal effect of design components: As seen in Table 3 (e.g., 2→7, 3→8, 4→9 comparisons), adding design components (e.g., backward layers, gating) only yields small incremental gains, which questions their relative contribution to the overall improvement.
> >
> > We would like to point out that the addition of key design components leads to significant performance gains, e.g., latent dynamics boosts mIoU by 2.1 (1 → 2) and refinement further boosts mIoU by 0.7 (4 → 5). We further observe that by removing design components like backward layers or multi-level, mIoU is reduced by 1.0 / 1.1 (6 → 7, 6 → 8), indicating the contribution of these components when used in tandem with the other components. Gating yields a 0.4 mIoU gain, and we further highlight how they prevent identity bias in the forward predictor in the bottom right of Figure 6 (red outlined box). Finally, we note that these performance improvements are in the ablation experimental setting and are expected to be larger in the full experimental setting, i.e., Midway Network obtains 39.7 mIoU in the full setting versus 31.5 in the ablation setting.
> >
> > > ### (Minor) Lack of planning or world-model evaluation: Despite the discussion of potential applications to planning and world modeling, no experiments demonstrate this capability, slightly limiting the perceived impact of the proposed approach.
> >
> > Yes, we are excited for this avenue of future work. However, we believe the impact of the proposed approach remains very strong as Midway Network introduces the first self-supervised learning method to jointly learn representations for recognition and motion solely from natural videos.
> >
> > > ### Why are optical flow results for PooDLe not reported in Tables 1–2? Was there a fundamental limitation in adapting its architecture for flow prediction, or were these experiments omitted for other reasons?
> >
> > Yes, PooDLe fundamentally incorporates an external flow network like RAFT for training, so to adapt its architecture for flow prediction would be to design a new architecture. This was also a motivating factor for Midway Network, which no longer requires this external dependency.

---

### Official Review · Reviewer_BVMm · 2025-10-27

**Soundness:** 3
**Presentation:** 3
**Contribution:** 3
**Rating:** 4
**Confidence:** 2

**Summary:**

This paper presents a novel self-supervised learning architecture for learning visual representations that jointly capture object recognition and motion understanding from video inputs. The proposed Midway Network achieves this by approximating a hierarchical latent embedding to model motion signals, thereby removing the need for an external motion predictor. Extensive experiments on object segmentation and optical flow tasks demonstrate the effectiveness of the approach.

**Strengths:**

1. By integrating latent optical flow estimation within a self-supervised framework, the proposed method enables simultaneous learning of motion information and content features through a single encoder. This design allows motion cues to be naturally incorporated into semantic representations.

2. The joint learning of motion and semantic features leads to consistent improvements in both semantic segmentation and optical flow performance.

**Weaknesses:**

1. While the paper contains several strong ideas, the overall framework is somewhat difficult to follow at first. It took multiple readings to fully grasp how the individual components contribute to the overall system. This could be improved by adding a concise, high-level overview of the architecture—perhaps at the beginning of Section 3—along with a summarizing figure that highlights the key components and their interactions.

2. Since one of the main contributions is the joint learning of motion and semantic representations from video, it would strengthen the paper to include a comparison with MC-JEPA to better contextualize performance gains.

**Questions:**

1. Figure 1 is referenced in the text but not included in the main manuscript.

2. Were all the additional components used during inference, or only in training?

---

> ### Author Response · Authors · 2025-11-21
> **Response to Reviewer BVMm**
>
> Thank you for taking the time to review our work and provide helpful feedback. We are glad you found that the paper has several strong ideas and that our proposed method leads to consistent improvement on both semantic segmentation and optical flow. We would like to address your comments below.
>
> > ### While the paper contains several strong ideas, the overall framework is somewhat difficult to follow at first. It took multiple readings to fully grasp how the individual components contribute to the overall system. This could be improved by adding a concise, high-level overview of the architecture—perhaps at the beginning of Section 3—along with a summarizing figure that highlights the key components and their interactions.
>
> We provide a high-level overview of Midway Network and how the primary components operate in the first paragraph of Section 3 on Page 4. This overview is accompanied by Figure 2, top of Page 4, which visually depicts the midway path, forward prediction blocks, backward layers, and hierarchical design. Could we clarify with you if you were looking for something beyond this overview?
>
> > ### Since one of the main contributions is the joint learning of motion and semantic representations from video, it would strengthen the paper to include a comparison with MC-JEPA to better contextualize performance gains.
>
> There are two reasons why we were unable to provide a comparison with MC-JEPA. First, to learn content features, MC-JEPA relies on VICReg applied to ImageNet, a curated iconic image dataset, whereas we target self-supervised learning on solely natural videos. Second, there is no public codebase available to reproduce the method nor adapt it to work with natural videos.
>
> > ### Figure 1 is referenced in the text but not included in the main manuscript.
>
> Figure 1 is on Page 2 of the main PDF. Could you please double-check on your end?
>
> > ### Were all the additional components used during inference, or only in training?
>
> For semantic segmentation, for linear readout only, the backward network is used following PooDLe. Otherwise, only the encoder is used for UperNet readout / finetuning. For optical flow estimation, we use the midway inverse and forward dynamics networks as a replacement for the binocular decoder when following CroCo v2’s setup.

---

### Official Review · Reviewer_RAZm · 2025-10-31

**Soundness:** 3
**Presentation:** 3
**Contribution:** 3
**Rating:** 6
**Confidence:** 2

**Summary:**

This paper introduces a new self-supervised learning framework that jointly learns object recognition and motion understanding from natural videos. It extends latent dynamics modeling to the video representation learning domain. The model combines an inverse-dynamics midway path, a dense forward prediction objective, and a hierarchical refinement structure to learn both semantic and motion features from unlabeled data.

**Strengths:**

1.The paper tackles an important and underexplored problem, learning both recognition and motion representations from unlabeled videos
2. It’s conceptually well-grounded, drawing inspiration from neuroscience to motivate the overall framework.
3. The architecture itself is Innovative. The midway path for motion latent inference feels like a fresh and thoughtful design choice.

**Weaknesses:**

1. It’s hard to disentangle architecture gains from data scale and model capacity
2. While the combination is novel, many components are incremental extensions of ideas from CroCo, DynaMo, or PooDLe.

**Questions:**

1. In Table 1, ViT-B only modestly improves flow over ViT-S and CroCo v2 still wins. What’s the failure mode for Midway at that scale?
2.The forwarded feature perturbation analysis is cool, could it be extended into a quantitative metric?

---

> ### Author Response · Authors · 2025-11-21
> **Response to Reviewer RAZm**
>
> Thank you for taking the time to review our work and provide valuable feedback. We are glad that you found that our paper tackles an important and underexplored problem, and that the architecture is conceptually well-grounded and innovative. We would like to address your comments below.
>
> > ### It’s hard to disentangle architecture gains from data scale and model capacity.
>
> Our paper isolates the architecture gains of Midway Network in three ways. First, we keep the pretraining data consistent across all of the baselines to ensure fair comparison between them. The pretraining datasets are also large, e.g., BDD100K is 778 hours and 70K instances of captured videos, and diverse across driving and city walking videos, to ensure that the architecture works well in larger data regimes. Second, we also use the same encoder architecture across all of the baselines and demonstrate that the improvements on semantic segmentation hold with and without the backward network. The same midway inverse and forward dynamics architectures are used for all baselines on optical flow evaluation, besides CroCo v2 which has its own binocular decoder. Third, we show that Midway Network continues to scale with larger model sizes from ViT-S to ViT-B in Table 1.
>
> > ### While the combination is novel, many components are incremental extensions of ideas from CroCo, DynaMo, or PooDLe.
>
> We would like to advocate that the novel combination is what allows Midway Network to tackle the joint problem of learning representations for both recognition and motion understanding whereas prior works only targeted one of the two. CroCo was trained using masked reconstruction to target stereo matching and optical flow, resulting in poorer semantic features for recognition. PooDLe had no mechanism for motion understanding, instead relying on an external optical flow network for pretraining. Finally, while DynaMo also uses inverse and forward dynamics for representation learning, Midway Network improves upon the latent dynamics architecture by introducing the hierarchical design of the midway path, where higher-level predictions are used to infer and refine lower-level latent dynamics. We also demonstrate the effectiveness of Midway Network on large-scale natural video datasets whereas DynaMo was only used in simulated / controlled environments.
>
> > ### In Table 1, ViT-B only modestly improves flow over ViT-S and CroCo v2 still wins. What’s the failure mode for Midway at that scale?
>
> Looking at a few examples, we observe that most of the improvements between ViT-S and ViT-B on flow for both Midway Network and CroCo v2 are fine-grained improvements on boundaries. Midway may be tailored towards prediction at the token-level because the prediction loss is on spatial tokens. However, overall, we observe minor differences in the optical flow estimates at that scale. We include these examples in **Appendix C, Figure 12** of the revised paper (green text).
>
> > ### The forwarded feature perturbation analysis is cool, could it be extended into a quantitative metric?
>
> Yes, we can evaluate the heatmaps produced by forwarded feature perturbation using optical flow maps. Because forwarded perturbation is on the token-level whereas optical flow maps are on the pixel-level, we sample the optical flow at the center of each token in the source frame and retrieve the target token that the optical flow maps to in the target frame (we also tried the optical flow averaged across the token with similar results). With this token-level optical flow, we compute the top-K accuracy of the forwarded perturbation heatmaps, i.e., for a source token, is the target token linked via optical flow within any of the top-K most similar target tokens in the heatmap? We use RAFT, a supervised-trained optical flow model, to compute optical flow maps for BDD. We then compare to heatmaps computed from cosine similarity of the last-layer dense features between the source and target frames of various baseline models. Table A below shows these quantitative results, where forwarded perturbation heatmaps are more well-aligned to the optical flow maps than feature similarity heatmaps from baseline models. The results are averaged over frame pairs, each randomly sampled from a random subset of 320 videos. We also provide visualizations of these heatmaps, converted to optical flow-like maps, compared to optical flow maps generated from RAFT in **Appendix D, Figure 15** of the revised paper (green text).
>
> | Method | Top-1 | Top-2 | Top-3 |
> | :--- | :--- | :--- | :--- |
> | Midway Network - BDD: Forwarded Perturbation | 73.93% | 88.20% | 92.39% |
> | Midway Network - BDD: Feature Similarity | 66.19% | 75.73% | 80.67% |
> | DINO - BDD: Feature Similarity | 58.59% | 68.66% | 74.27% |
> | DINO - ImageNet: Feature Similarity | 55.29% | 66.21% | 71.89% |
>
> Table A. Top-K accuracy of forwarded feature perturbation and feature similarity heatmaps evaluated against optical flow maps predicted by RAFT for BDD frame pairs.

---

### Meta-Review · Area_Chair_VcXd · 2026-01-04

**Summary:**

Across reviewers, the common concerns center on limited technical novelty, insufficient theoretical grounding, and gaps in experimental validation.

a) While the proposed framework combines multiple ideas in a coherent way, many reviewers note that its core components are largely incremental extensions of prior work (e.g., CroCo, PooDLe, latent world models), and the individual design elements contribute only modest gains. Several reviewers question whether the learned latent representations truly encode motion rather than implicitly leaking target information, noting the lack of formal guarantees and the reliance on largely qualitative analyses (e.g., forward feature perturbation). More broadly, the theoretical link between latent dynamics, motion prediction, and semantic invariance remains intuitive but underdeveloped.

b) On the experimental side, reviewers consistently point out limited quantitative validation: motion learning and interpretability are not rigorously measured against ground truth, ablations suggest marginal effects of added complexity, comparisons to key baselines (e.g., MC-JEPA, PooDLe flow results) are incomplete, and evaluations are restricted in scale, diversity, temporal horizon, and statistical reporting (e.g., missing multi-step prediction, broader pretraining data, and variance across seeds).

**Reviewer Concerns:**

The authors provided additional evaluations related to motion learning, including comparisons between forward-perturbation–based feature similarity heatmaps and pseudo motion ground truth. These additions partially address reviewers’ concerns regarding the lack of rigorous evaluation and theoretical justification for whether the learned representations genuinely encode motion.
However, several concerns remain only partially resolved, notably the limited novelty of applying inverse + forward dynamics framework for learning latent motions, the marginal performance gains of individual design components relative to the added architectural complexity, as well as the framework’s ability to capture long-term temporal correlations and multi-step dynamics.

**Reviewer Scores:**

(a) Reviewer RAZm would likely maintain their original score, as they did not appear to be fully engaged beyond their initial review.
(b) Reviewer BVMm would likely maintain their original score, as they did not appear to be fully engaged beyond their initial review.
(c) Reviewer 9dmU would likely maintain their original score, since their main concerns pertain to technical novelty and modest performance gain, which was only partially addressed in the rebuttal.
(d) Reviewer m9iU actively participated in the discussion and would likely maintain their original score of 6, as they clearly articulated which concerns were adequately addressed by the rebuttal and which remained insufficiently resolved.

---

### Decision · Program_Chairs · 2026-01-26

Accept (Poster)